# UNBALANCED DIFFUSION SCHRÖDINGER BRIDGE

## ABSTRACT

*Schrödinger bridges* (SBs) provide an elegant framework for modeling the temporal evolution of populations in physical, chemical, or biological systems. Such natural processes are commonly subject to changes in population size over time due to the emergence of new species or birth and death events. However, existing neural parameterizations of SBs such as *diffusion Schrödinger bridges* (DSBs) are restricted to settings in which the endpoints of the stochastic process are both *probability measures* and assume *conservation of mass* constraints. To address this limitation, we introduce *unbalanced* DSBs which model the temporal evolution of marginals with arbitrary finite mass. This is achieved by deriving the time reversal of stochastic differential equations with *killing* and *birth* terms. We introduce two novel algorithmic approaches that incorporate a scalable objective function designed for training imbalanced DSBs. Additionally, we showcase their robustness through challenging applications, including predicting diverse molecular single-cell responses to cancer drugs and simulating the emergence and dissemination of novel viral variants.

## 1 INTRODUCTION

Modeling the evolution of probability distributions is a foundational task that finds wide-ranging applications in diverse fields, including the natural sciences (Bunne et al., 2021; Schiebinger et al., 2019), signal processing (Kolouri et al., 2017), and economics (Galichon, 2018). In these domains, observations are often limited to discrete time points, making it challenging to track the trajectories of individual elements over time.

To address this challenge, the *Schrödinger bridge*, also known as dynamic entropic optimal transport (Santambrogio, 2015; Villani, 2009; Vargas et al., 2021b), has recently emerged as a powerful tool for reconstructing population dynamics from periodic snapshots (De Bortoli et al., 2021; Chen et al., 2021). The key insight is that Schrödinger bridge can be efficiently solved using the so-called *diffusion Schrödinger bridge* (DSB) framework, which consists of iteratively applying *score matching* techniques inspired by diffusion models (Song et al., 2021b). This close connection with diffusion modeling, a technique renowned for its remarkable success in fields such as image and language processing, has greatly enhanced the practical utility of DSBs.

Despite their impressive successes, one major limitation of SB approaches is their inherent assumption of *mass conservation*. While valid in many scenarios, this assumption fails to account for birth and death phenomena critical in biological and medical applications (see Fig. 1 for an illustration). For example, assessing cellular responses to cancer drugs must account for cell death, leading to unequal cell counts before and after treatment. Similarly, tracking the dynamics of infectious diseases necessitates considering virus proliferation and death.

Motivated by these applications, our paper aims to extend the DSB framework to accommodate distributions with *unequal* mass. Specifically, when given two measures $\mu_0$ and $\mu_1$ such that $\mu_0(\mathbb{R}^d) \neq \mu_1(\mathbb{R}^d)$, our goal is to find a dynamics $\mu_t$ of *non-probability* measures that interpolates between $\mu_0$ and $\mu_1$. Existing DSB methods cannot be straightforwardly applied in this unbalanced setting due to the following challenges:

- Theoretically, handling unbalanced Schrödinger bridge corresponds to extending the DSB framework from $\mathbb{R}^d$ to its *one-point compactification* $\mathbb{R}^d \cup \{\infty\}$, where $\infty$ acts as a "coffin state". Achieving this extension requires non-trivial adaptations of the DSB framework based on generator theory.

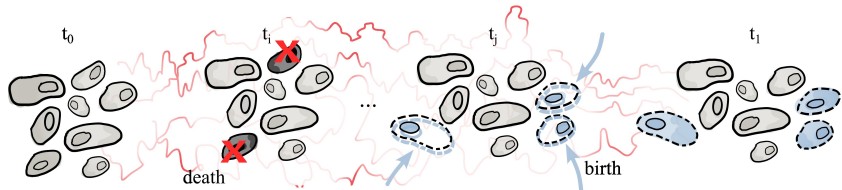

Figure 1: *Unbalanced* Schrödinger bridge. The time evolution of a cell population encompasses alterations in transcriptomic profiles as well as natural and externally induced death and birth events.

- Empirically, much of the strength of DSB stems from the score-matching framework. However, in the unbalanced setting, score matching alone turns out to be *insufficient*. Instead, an additional density estimation procedure is required, introducing considerable computational complexity.

**Our Approach and Contributions.** To address the above challenges, we exploit the insightful perspective of *time-reversal* within the context of score matching. Specifically, we first recall that score-matching can be seen as time-reversing *stochastic differential equations* (SDEs) on $\mathbb{R}^d$ (Song et al., 2021b), a fundamental concept that also underpins DSB schemes. The centerpiece of our framework is to generalize this procedure to derive the time-reversal formulas for SDEs on $\mathbb{R}^d \cup \{\infty\}$. A crucial outcome of our theory is proving that time-reversal of an SDE with *death* (diffusions with jumps *into* the coffin state $\infty$) is another SDE with *birth* (diffusions with jumps *out of* the coffin state $\infty$). Conversely, we show that the time-reversal of an SDE with birth is a SDE with death. This duality has the important implication that, to solve unbalanced Schrödinger bridge numerically, it suffices to design algorithms for time-reversing SDEs with birth and death, respectively. Armed with this insight, we then design scalable algorithms to address the unbalanced DSBs problems.

In summary, our contributions include:

1. Showing that the time-reversal of diffusions with death correspond to diffusions with birth and, similarly, that time-reversing diffusions with birth yields diffusions with death.
2. Leveraging these formulas and associated loss functions to propose a framework for solving the DSB on $\mathbb{R}^d \cup \{\infty\}$. We also design scalable algorithms to approximately tackle these unbalanced DSBs problems with both deaths and births.
3. Measuring the performance of our algorithms on various tasks, with a particular focus on enhancing the modeling of cellular responses to cancer drugs. Notably, our experiments demonstrate a significant impact of including information on deaths and births occurring at intermediate times.

## 2 BACKGROUND

**Diffusion Schrödinger bridge.** The *Schrödinger bridge* (SB) problem refers to the following KL-minimization problem with constrained marginals at both the initial and final time points:

$$\mathbb{P}^{\mathrm{SB}} \coloneqq \mathrm{argmin}\{\mathrm{KL}(\mathbb{P}|\mathbb{P}^0) \, : \, \mathbb{P}_0 = \mu_0, \ \mathbb{P}_T = \mu_1\}, \tag{1}$$

where $\mu_0, \mu_1$ are probability measures and $\mathbb{P}^0$ is a reference probability path measure, i.e. a probability measure on the space of continuous functions $\mathrm{C}([0,T], \mathbb{R}^d)$. A conceptual algorithm for solving (1) is the iterative proportional fitting (IPF) iterations (Sinkhorn and Knopp, 1967; Knight, 2008; Peyré and Cuturi, 2019; Cuturi and Doucet, 2014) which define processes $(\mathbb{P}^n)_{n \in \mathbb{N}}$ via

$$\mathbb{P}^{2n+1} = \mathrm{argmin}\{\mathrm{KL}(\mathbb{P}|\mathbb{P}^{2n}) \, : \, \mathbb{P}_T = \mu_1\}, \quad \mathbb{P}^{2n+2} = \mathrm{argmin}\{\mathrm{KL}(\mathbb{P}|\mathbb{P}^{2n+1}) \, : \, \mathbb{P}_0 = \mu_0\}. \tag{2}$$

Under mild assumptions, it can be shown that $\lim_{n \to +\infty} \mathbb{P}^n = \mathbb{P}^{\mathrm{SB}}$.

While conceptually appealing, the IPF algorithm is nonetheless not practically implementable since it operates on path measures. To address this limitation and numerically approximate solutions to (1), De Bortoli et al. (2021) introduced the *diffusion Schrödinger bridge* (DSB) framework, which is rooted in the following key observation: Assuming that $\mathbb{P}^n$ is associated with the SDE $\mathrm{d}\mathbf{X}_t = f_t(\mathbf{X}_t)\mathrm{d}t + \mathrm{d}\mathbf{B}_t$, then $\mathbb{P}^{n+1}$ is associated with the *time-reversal* (Haussmann and Pardoux,

1986; Anderson, 1982; Cattiaux et al., 2021) of this SDE, initialized at $\mu_1$. More precisely, we have $(\mathbf{Y}_{T-t})_{t \in [0,T]} \sim \mathbb{P}^{n+1}$ with

$$\mathrm{d}\mathbf{Y}_t = [-f_{T-t}(\mathbf{Y}_t) + \nabla \log p_{T-t}(\mathbf{Y}_t)]\mathrm{d}t + \mathrm{d}\mathbf{B}_t, \qquad \mathbf{Y}_0 \sim \mu_1$$

where $p_t$ is the density of $\mathbb{P}_t^n$ and $(\mathbf{B}_t)_{t \geq 0}$ a $d$-dimensional Brownian motion. The quantity $\nabla \log p_t$ is called the *Stein score* and can be estimated using score-matching techniques (Hyvärinen and Dayan, 2005; Vincent, 2011) and the powerful diffusion models (Song et al., 2021b; Ho et al., 2020).

**Unbalanced Schrödinger bridge and optimal transport.** This paper focuses on solving the *unbalanced* Schrödinger bridge problem, which arises when the initial and terminal measures do not have the same total mass, meaning $\mu_0(\mathbb{R}^d) \neq \mu_1(\mathbb{R}^d)$ in (1). Additionally, $\mathbb{P}^0$ is a reference *non-probability* path measure that might not have a total mass of 1 at each time point.

Existing work already addresses the unbalanced SB and related unbalanced optimal transport (OT) problems. Some approaches relax the classical OT formulation by introducing "soft" marginal constraints instead of hard ones (Chizat et al., 2018; Liero et al., 2018; Yang and Uhler, 2019; Kondratyev et al., 2016). Other methods extend finite measures to measures with the same mass by incorporating a "coffin" state and performing optimal transport on an extended space (Pele and Werman, 2009; Caffarelli and McCann, 2010; Gramfort et al., 2015; Ekeland, 2010). Finally, some theoretical properties of the solutions of unbalanced SB, such as asymptotic convergence or connection to stochastic optimal control, are established in (Chen et al., 2022).

To the best of our knowledge, however, our work is the first to derive the *time-reversal formula* for the unbalanced SB problem, which is crucial for establishing a connection to diffusion models and thereby taking advantage of their empirical performance.

## 3 TIME-REVERSAL OF BIRTH AND DEATH PROCESSES

To numerically address the unbalanced Schrödinger bridge problem, it is crucial to derive formulas for the IPF procedure which can be easily discretized. In particular, following the framework of De Bortoli et al. (2021), we need to derive time-reversal formulas. In this section, we present a succinct overview of the core concepts and leave a thorough discussion to Appendices A to D.

**Diffusion processes with killing.** A diffusion process with drift $b(\cdot)$ and *killing* rate $k(\cdot)$ can be intuitively understood in terms of its SDE representation:

$$\mathbf{X}_{t+\mathrm{d}t} = \begin{cases} \infty, & \text{with probability } k(\mathbf{X}_t)\mathrm{d}t \\ \mathbf{X}_t + b(\mathbf{X}_t)\mathrm{d}t + \mathrm{d}\mathbf{B}_t, & \text{with probability } 1 - k(\mathbf{X}_t)\mathrm{d}t, \end{cases} \tag{3}$$

where $\infty$ represents the *coffin* state after a particle is killed. In this coffin state, a particle ceases to move further: $X_{t+s} = \infty$ for all $s \geq 0$ if $X_t = \infty$. To lighten our notation, we omit the possible dependency of $k$ on time.

Despite intuitively appealing, (3) is mathematically ill-defined and hence not suitable for developing a rigorous framework. Instead, we opt for the *generator* framework as follows. Denote the *one-point compactification* of $\mathbb{R}^d$ by $\hat{\mathbb{R}}^d = \mathbb{R}^d \cup \{\infty\}$. For a smooth function $f$ on $\hat{\mathbb{R}}^d$, we define an operator $\hat{\mathcal{K}}$ as

$$\hat{\mathcal{K}}(f)(x) = \left[ \langle b(x), \nabla f(x) \rangle + \tfrac{1}{2}\Delta f(x) - k(x)(f(x) - f(\infty)) \right] \mathbf{1}_{\mathbb{R}^d}(x). \tag{4}$$

It can be shown that (4) is the *infinitesimal generator* associated with (3), i.e., $\hat{\mathcal{K}}(f)(x) = \lim_{t \to 0}(1/t)(\mathbb{E}[f(\mathbf{X}_t) \mid \mathbf{X}_0 = x] - f(x))$ for a process $(\mathbf{X}_t)_{t \geq 0}$ which is a diffusion with killing rate $k(\cdot)$ on $\hat{\mathbb{R}}^d$. To get a sense of why the generator is a useful notion, one can informally replace $f(\cdot)$ with $\mathbf{1}_y(\cdot)$ for an arbitrary $y \in \hat{\mathbb{R}}^d$, $y \neq x$ and note that the previous expression simplifies to $\hat{\mathcal{K}}(f)(x) = \lim_{t \to 0}(1/t)\mathbb{P}(\mathbf{X}_t = y | \mathbf{X}_0 = x)$. Therefore, the knowledge of $\hat{\mathcal{K}}(f)$ on all functions $f$ completely characterizes the infinitesimal change of the probability transition kernel of the diffusion. A rigorous treatment of such processes is provided in Proposition C.6.

**Time-reversal of diffusions with killing and birth.** We are now ready to derive our main result: a time-reversal formula for diffusion processes with killing. Let $(\mathbf{X}_t)_{t \in [0,T]}$ be the diffusion process with killing rate $k$, defined by the generator (4), and $(\mathbf{Y}_t)_{t \in [0,T]} := (\mathbf{X}_{T-t})_{t \in [0,T]}$ be its *time reversal*.

**Proposition 3.1** (Time reversal; informal). *Under mild assumptions, the generator $\hat{\mathcal{B}}$ of the time-reversed process $(\mathbf{Y}_t)_{t\in[0,T]}$ is given for any sufficiently smooth function $f$ and any $x \in \hat{\mathbb{R}}^d$ by*

$$\hat{\mathcal{B}}(f)(t,x) = \left[ \langle -b(x) + \nabla \log p_{T-t}(x), \nabla f(x) \rangle + \tfrac{1}{2}\Delta f(x) \right] \mathbf{1}_{\mathbb{R}^d}(x) \tag{5}$$
$$+ \int_{\mathbb{R}^d}(p_{T-t}(\tilde{x})/S_{T-t})k(\tilde{x})(f(\tilde{x}) - f(\infty))\mathrm{d}\tilde{x}\, \mathbf{1}_{\infty}(x).$$

*Here, $p_t$ is the probability density of $\mathbf{X}_t$ and $S_t := \mathbb{P}[\mathbf{X}_t = \infty]$.*

The key terms appearing in (5) are highlighted in blue. Note that, if there is no killing in $\mathbf{X}_t$, i.e., $k(x) \equiv 0$, then $\hat{\mathcal{B}}(f)(t,x)$ reduces to the well-known time-reversal formula for Euclidean diffusions (Haussmann and Pardoux, 1986; Cattiaux et al., 2021). In the presence of killing, instead, Proposition 3.1 leads to the important observation that *the time-reversal of a diffusion process with killing is a diffusion process with **birth***: Since (5) traces the behavior of the process (3) backward in time, it appears as though a particle "revives" from the coffin state $\infty$, and this revival occurs at a certain (time-dependent) birth rate denoted by $q_t(x)$. Subsequently, the particle proceeds along the reverse SDE *without ever re-entering the coffin state*. Importantly, Proposition 3.1 furnishes an explicit formula for the birth rate: $q_t(x) = k(x)p_t(x)/S_t$.

Analogously, we can derive the time reversal of a diffusion process with *birth* in a similar fashion to that of Proposition 3.1: Consider a diffusion process given by (5), i.e., with drift $b(\cdot)$ and *birth rate* $q_t(x) := k(x)p_t(x)/S_t$. Then, under mild assumptions, one can show that the generator $\hat{\mathcal{K}}$ associated with the time-reversed diffusion with birth in (5) can be expressed as

$$\hat{\mathcal{K}}(f)(t,x) = \left[ \langle -b(x) + \nabla \log p_{T-t}(x), \nabla f(x) \rangle + \tfrac{1}{2}\Delta f(x) \right] \mathbf{1}_{\mathbb{R}^d}(x) \tag{6}$$
$$- (S_{T-t}/p_{T-t}(x))q_{T-t}(x)(f(x) - f(\infty))\, \mathbf{1}_{\mathbb{R}^d}(x)$$

for any sufficiently smooth function $f$ and $x \in \hat{\mathbb{R}}^d$. The key terms in (6) are highlighted in red. Comparing (6) with (4), we conclude that the time-reversal of a diffusion with birth rate $q_t(x)$ is a diffusion with killing rate $k_t(x) := S_t q_t(x)/p_t(x)$.

## 4 UNBALANCED ITERATIVE PROPORTIONAL FITTING

Equipped with the notion of time reversal, we are now ready to derive an IPF scheme for marginals with arbitrary mass. Our approach begins by deriving the optimality conditions for the unbalanced SB, which will serve as the basis for unveiling the corresponding IPF scheme.

**Unbalanced Schrödinger bridge: Optimality conditions.** For simplicity, we assume that the target measures $\mu_0, \mu_1$ satisfy $\mu_0(\mathbb{R}^d) \geq \mu_1(\mathbb{R}^d)$ and their restrictions to $\mathbb{R}^d$ have smooth densities $p_0, p_1$ w.r.t. the Lebesgue measure. We start by characterizing the *optimality condition* of the solution to the unbalanced SB objective in (1). To this end, we need the following extension of the *Schrödinger equations* to unbalanced marginals (Chen et al., 2022): Let the process with infinitesimal generator $\hat{\mathcal{K}}^0$ defined in (4) be our prior measure $\mathbb{P}^0$. Unlike in Section 2, we consider $\mathbb{P}^{\mathrm{SB}}$ and $\mathbb{P}^0$ to be path measures on the *extended* space $\hat{\mathbb{R}}^d$. To state the solution to (1) in this case, we define the functions $(\varphi, \hat{\varphi}, \Psi, \hat{\Psi})$ which, for any $t \in [0,T]$ and $x \in \mathbb{R}^d$, satisfy:

$$\partial_t \varphi_t(x) = -\langle b(x), \nabla \varphi_t(x) \rangle - \tfrac{1}{2}\Delta\varphi_t(x) + k(x)\varphi_t(x) - k(x)\Psi_t, \quad \partial_t \Psi_t = 0, \tag{7}$$

$$\partial_t \hat{\varphi}_t(x) = -\mathrm{div}(b\hat{\varphi}_t)(x) + \tfrac{1}{2}\Delta\hat{\varphi}_t(x) - k(x)\hat{\varphi}_t(x), \qquad \partial_t \hat{\Psi}_t = \int_{\mathbb{R}^d} k(x)\hat{\varphi}_t(x)\mathrm{d}x,$$

together with the boundary conditions:

$$\varphi_0\hat{\varphi}_0 = p_0, \quad \varphi_1\hat{\varphi}_1 = p_1 \quad \text{and} \quad \Psi_0\hat{\Psi}_0 = 0, \quad \Psi_1\hat{\Psi}_1 = \mu_0(\mathbb{R}^d) - \mu_1(\mathbb{R}^d).$$

To improve readability, we highlight the control terms $(\varphi, \Psi)$ of the process with killing in red and those of the process with birth in blue. The proof of the following result is given in Appendix C.

**Proposition 4.1.** *Under mild conditions, there exists a unique solution $\mathbb{P}^{\mathrm{SB}}$ to (1) on $\hat{\mathbb{R}}^d$. The paths $(\mathbf{X}_t)_{t\in[0,T]} \sim \mathbb{P}^{\mathrm{SB}}$ are associated with the generator $\hat{\mathcal{K}}^{\mathrm{SB}}$ given, for any smooth $f$, $t \in [0,T]$ and $x \in \hat{\mathbb{R}}^d$, by*

$$\hat{\mathcal{K}}_t^{\mathrm{SB}}(f)(t,x) = \left[ \langle b(x) + \nabla \log \varphi_t(x), \nabla f(x) \rangle + \tfrac{1}{2}\Delta f(x) - (\Psi_t/\varphi_t(x))k(x)(f(x) - f(\infty)) \right] \mathbf{1}_{\mathbb{R}^d}(x). \tag{8}$$

*A similar result can be derived for the birth process $(\mathbf{X}_{T-t})_{t\in[0,T]}$.*

Proposition 4.1 reveals that, when the reference is a killing process, the solution $\mathbb{P}^{\mathrm{SB}}$ is *also a killing process*. In particular, (8) implies that the difference between its generator and the one of the reference depends only on $(\varphi, \Psi)$. Analogously, the time reversal of $\mathbb{P}^{\mathrm{SB}}$ is a birth process which only depends on $(\hat{\varphi}, \hat{\Psi})$.

**Unbalanced Schrödinger bridge: Iterative proportional fitting.** We are now ready to describe the unbalanced IPF scheme which serves as the foundation for our proposed algorithms. We consider the sequence $(\mathbb{P}^n)_{n \in \mathbb{N}}$ of measures on $\hat{\mathbb{R}}^d$, given by (2), and recall that $\mathbb{P}^0$ is associated with a diffusion *with killing*. For any $n \in \mathbb{N}$, we have that: (a) $\mathbb{P}^{2n+1}$ is a diffusion *with birth*, given by the time-reversal of $\mathbb{P}^{2n}$, initialized at $\mu_1$; (b) $\mathbb{P}^{2n+2}$ is a diffusion *with killing*, given by the time-reversal of $\mathbb{P}^{2n+1}$, initialized at $\mu_0$. In order to extend DSBs to *unbalanced* problems, we need to estimate the parameters appearing in (5) and (6). Contrary to De Bortoli et al. (2021), it is not sufficient to estimate solely the drift of the diffusion, since we also need to update the killing/birth rate. The exact update formulas is given in the following result.

**Proposition 4.2.** *Under mild assumptions, there exist $(\varphi^n, \hat{\varphi}^n, \Psi^n, \hat{\Psi}^n)_{n \in \mathbb{N}}$ such that, for any $n \in \mathbb{N}$, $\mathbb{P}^{2n}$ is initialized at $\mathbb{P}_0^{2n} = \mu_0$ and associated with $\hat{\mathcal{K}}^n$ given for any $t \in [0, T]$ and $x \in \hat{\mathbb{R}}^d$ by*

$$\hat{\mathcal{K}}_t^n(f)(t, x) = \left[ \langle b(x) + \nabla \log \varphi_t^n(x), \nabla f(x) \rangle + \tfrac{1}{2} \Delta f(x) - k(x)(\Psi_t^n / \varphi_t^n(x))(f(x) - f(\infty)) \right] \mathbf{1}_{\mathbb{R}^d}(x).$$

*Similarly, $\mathbb{P}^{2n+1}$ is initialized at $\mathbb{P}_T^{2n+1} = \mu_1$ and associated with $\hat{\mathcal{B}}^{n+1}$ given by*

$$\hat{\mathcal{B}}_t^{n+1}(f)(t, x) = [\langle -b(x) + \nabla \log \hat{\varphi}_{T-t}^{n+1}(x), \nabla f(x) \rangle + \tfrac{1}{2} \Delta f(x)] \mathbf{1}_{\mathbb{R}^d}(x) \qquad (9)$$
$$+ \int_{\mathbb{R}^d} (\hat{\varphi}_{T-t}^{n+1}(\tilde{x}) / \hat{\Psi}_{T-t}^{n+1}) k(\tilde{x})(f(\tilde{x}) - f(\infty)) \mathrm{d}\tilde{x} \mathbf{1}_\infty(x).$$

*Furthermore, we have that for, any $n \in \mathbb{N}$, $t \in [0, T]$ and $x \in \mathbb{R}^d$,*

$$\log \varphi_t^n(x) + \log \hat{\varphi}_t^{n+1}(x) = \log p_t^{2n}(x), \qquad \Psi_t^n \hat{\Psi}_t^{n+1} = 1 - \int_{\mathbb{R}^d} p_t^{2n}(x) \mathrm{d}x,$$
$$\log \varphi_t^{n+1}(x) + \log \hat{\varphi}_t^{n+1}(x) = \log p_t^{2n+1}(x), \qquad \Psi_t^{n+1} \hat{\Psi}_t^{n+1} = 1 - \int_{\mathbb{R}^d} p_t^{2n+1}(x) \mathrm{d}x, \qquad (10)$$

*where $p_t^n$ is the density w.r.t. the Lebesgue measure of $\mathbb{P}_t^n$ restricted to $\mathbb{R}^d$.*

The crucial takeaway from Proposition 4.2 is that the unbalanced IPF iterations demand updates to the functions $(\varphi^n, \hat{\varphi}^n, \Psi^n, \hat{\Psi}^n)_{n \in \mathbb{N}}$. As indicated by (10), these updates in turns require to estimate the *density* $(p_t^n)_{n \in \mathbb{N}}$. This is in contrast to the balanced setting where only the *score* $\nabla \log p_t^n$ is required.

**Sampling from diffusion with killing or birth.** Finally, whether it be to approximate the quantity $(\varphi^n, \hat{\varphi}^n, \Psi^n, \hat{\Psi}^n)_{n \in \mathbb{N}}$ or to sample from the (approximate) unbalanced Schrödinger bridge, we need to discretize birth and death dynamics. We start by discussing how to sample from a diffusion with killing of the form (4). Our approach is based on the Lie-Trotter-Kato formula (Ethier and Kurtz, 2009, Corollary 6.7, p.33). Since $\hat{\mathcal{K}}$, given by (4), can be decomposed in $\hat{\mathcal{K}}_{\mathrm{cont}}$ and $\hat{\mathcal{K}}_{\mathrm{disc}}$ such that

$$\hat{\mathcal{K}}_{\mathrm{cont}}(f)(x) = \langle b(x), \nabla f(x) \rangle + \tfrac{1}{2} \Delta f(x), \qquad \hat{\mathcal{K}}_{\mathrm{disc}}(f)(x) = -k(x)(f(x) - f(\infty)),$$

we alternate between sampling according to $\hat{\mathcal{K}}_{\mathrm{cont}}$ and to $\hat{\mathcal{K}}_{\mathrm{disc}}$, where the latter describes a *pure* killing process. For small-enough step-sizes, Ethier and Kurtz (2009, Corollary 6.7, p.33) asserts that this amounts to sampling according to $\hat{\mathcal{K}}$. We discretize the diffusion process using the standard Euler-Maruyama method and use coin flipping for $\hat{\mathcal{K}}_{\mathrm{disc}}$ (see Appendix D for details).

Sampling diffusion with birth is more problematic. We can still decompose $\hat{\mathcal{B}}$ from (5) into $\hat{\mathcal{B}}_{\mathrm{cont}}$ and $\hat{\mathcal{B}}_{\mathrm{disc}}$ with

$$\hat{\mathcal{B}}_{\mathrm{cont}}(f)(x) = \langle b(x), \nabla f(x) \rangle + \tfrac{1}{2} \Delta f(x), \qquad \hat{\mathcal{B}}_{\mathrm{disc}}(f)(\infty) = \int_{\mathbb{R}^d} q(\tilde{x})(f(\tilde{x}) - f(\infty)) \mathrm{d}\tilde{x},$$

but $\hat{\mathcal{B}}_{\mathrm{disc}}$ is a *pure* birth process, which is more difficult to sample from. We achieve the task by considering time intervals of size $\gamma > 0$ and assuming that the birth rate can be written as $q(\tilde{x}) = \ell(\tilde{x}) p(\tilde{x})$, where $\int_{\mathbb{R}^d} p(\tilde{x}) \mathrm{d}\tilde{x} = 1$. We can then sample $\hat{X} \sim p$ and then let $X = \hat{X}$ with probability $\ell(\hat{X}) \gamma$ and $X = \infty$ otherwise, in which case the birth is rejected.

When executing our IPF scheme, the form of $\hat{\mathcal{B}}_{\mathrm{disc}}^{n+1}$, given by (9), is

$$\hat{\mathcal{B}}_{\mathrm{disc}}^{n+1}(f)(t,\infty) = \int_{\mathbb{R}^d} (\hat{\varphi}_{T-t}^{n+1}(\tilde{x})/\hat{\Psi}_{T-t}^{n+1})k(\tilde{x})(f(\tilde{x})-f(\infty))\mathrm{d}\tilde{x}$$
$$= \int_{\mathbb{R}^d} p_{T-t}^{2n}(\tilde{x})k(\tilde{x})/(\varphi_{T-t}^n(x)\hat{\Psi}_{T-t}^{n+1})(f(\tilde{x})-f(\infty))\mathrm{d}\tilde{x},$$

where we obtained the second equality using (10). We therefore have that $\ell(\tilde{x}) = k(\tilde{x})/(\varphi_{T-t}^n(x)\hat{\Psi}_{T-t}^{n+1})$ and $p(\tilde{x}) = p_{T-t}^{2n}(\tilde{x})$. To sample backward paths $\mathbb{P}^{2n+1}$, we first sample a forward trajectory $(\mathbf{X}_t)_{t\in[0,T]} \sim \mathbb{P}^{2n}$. If the particle is at $\infty$ at time $t$, then we sample a Bernoulli variable with parameter $\gamma k(\mathbf{X}_{T-t})/(\varphi_{T-t}^n(\mathbf{X}_{T-t})\hat{\Psi}_{T-t}^{n+1})$. The particle is born if this random variable is equal to one. In that case, we set $\mathbf{Y}_{t+\gamma} = \mathbf{X}_{T-t}$ (and $\mathbf{Y}_{t+\gamma} = \infty$ otherwise). After birth, backward trajectories evolve according to the continuous diffusion in $\mathbb{R}^d$ described by $\hat{\mathcal{B}}_{\mathrm{cont}}$. We call this method *shadow trajectory sampling* and illustrate it in Fig. 2 while providing the algorithm in Appendix D.

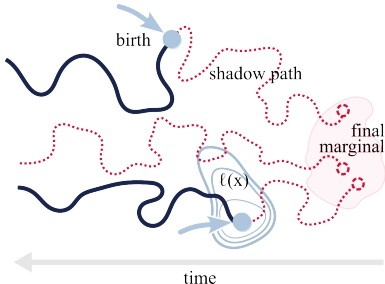

Figure 2: Shadow trajectory sampling.

# 5 UNBALANCED DIFFUSION SCHRÖDINGER BRIDGE WITH TEMPORAL DIFFERENCE LOSS

Equipped with Proposition 4.2 and the sampling strategies presented in the previous paragraph, we are now ready to describe our numerical implementation of (2). We parameterize $\log \varphi_t^n, \log \hat{\varphi}_t^n$ with the two networks $f_{t,\theta_n}$ and $\hat{f}_{t,\hat{\theta}_n}$, respectively. Note that, contrary to the classical diffusion Schrödinger bridge setting (De Bortoli et al., 2021; Chen et al., 2021), we need to estimate $\log \varphi_t^n, \log \hat{\varphi}_t^n$ and not only $\nabla \log \varphi_t^n, \nabla \log \hat{\varphi}_t^n$.

**Estimating** $\log \varphi^n$ **and** $\log \hat{\varphi}^n$. To learn $\hat{f}_{t,\hat{\theta}^{n+1}}$, we first sample from $\mathbb{P}^{2n}$ using $f_{t,\theta^n}$ and $\Psi_t^n$, which are assumed to be known. Once this is done, we compute a loss on $\hat{f}_{t,\hat{\theta}^{n+1}}$ combining the *mean-matching* (MM) loss (De Bortoli et al., 2021) (which is a loss on $\nabla \hat{f}_{t,\hat{\theta}}$) and a *temporal difference* (TD) loss (Liu et al., 2022) (which is a loss on $\hat{f}_{t,\hat{\theta}}$). The MM loss was designed to compute iterates in classical DSB while the TD loss was introduced in Liu et al. (2022) in order to compute *generalized* Schrödinger bridges. The TD loss can be seen as a *regularizer* to the MM loss. Indeed, at equilibrium, minimizers $\hat{f}_{t,\hat{\theta}^\star}$ of the MM loss satisfy $\hat{f}_{t,\hat{\theta}^\star} = \log \hat{\varphi}_t^\star + c_t$, where $c_t$ only depends on the time $t$. In Appendix F, we provide further details on how the MM and TD losses can be adapted to the *unbalanced* setting. So far we have described how to update $\hat{f}_{t,\theta^{n+1}}$ given $f_{t,\theta^n}$ and $\Psi_t^n$. The update of $\hat{\Psi}_t^{n+1}$, instead, leverages the closed-form expression $\hat{\Psi}_t^{n+1} = (1 - \int_{\mathbb{R}^d} p_t^{2n}(x)\mathrm{d}x)/\Psi_t^n$, which follows from (10). $\hat{\Psi}_t^{n+1}$ can, in fact, be computed on the fly, by approximating $\int_{\mathbb{R}^d} p_t^{2n}(x)\mathrm{d}x$ with the proportion of *live* particles at time $t$ in the forward process. Once we have estimated $(\hat{f}_{t,\hat{\theta}^{n+1}}, \hat{\Psi}^{n+1})$ we can estimate $(f_{t,\theta^{n+1}}, \Psi_t^{n+1})$ in a similar fashion.

**Updating** $\Psi^n$. We know from (7) that at $\Psi_t^n$ does not depend on $t$.[1] Using this observation, we consider a *correction* strategy which projects $\Psi_t$ on constant functions. As a consequence of (7), we have for any $n \in \mathbb{N}$,

$$\Psi^n = (\mu_0(\mathbb{R}^d) - \mu_1(\mathbb{R}^d))/\int_0^1 \int_{\mathbb{R}^d} k(\tilde{x})\hat{\varphi}_t^n(\tilde{x})\mathrm{d}\tilde{x}. \tag{11}$$

We propose a numerical approximation of (11) in Appendix F.

**Algorithm.** We are now ready to introduce our numerical approximation of Unbalanced IPF, termed *Unbalanced Diffusion Schrödinger Bridge with Temporal Difference* (UDSB-TD) and described in Algorithm 1. The procedures UPDATE-PSI, SAMPLE-FORWARD and SAMPLE-BACKWARD are given

---

[1]This is because the system (7) is still valid for the *iterates* $(\varphi^n, \hat{\varphi}^n, \Psi^n, \hat{\Psi}^n)_{n\in\mathbb{N}}$.

in Appendix F. While our algorithm resembles the one of Liu et al. (2022), we highlight some key differences: (a) In Liu et al. (2022), the forward and backward processes do not incorporate killing and/or birth; (b) as a consequence, it is not possible to update the killing and birth rates to match a desired mass constraint. It can, in fact, be shown that the formulation of Liu et al. (2022) corresponds to the *reweighted* approach in Chen et al. (2022), which is *not* equivalent to the unbalanced SB.

---

**Algorithm 1** UDSB-TD training

---

**Input:** $\theta, \hat{\theta}, \mu_0, \mu_1$
**Output:** $\theta, \hat{\theta}$
1: **for** epoch $n \in \{0, ..., N\}$ **do**
2:     $\psi \leftarrow$ UPDATE-PSI$(\theta, \hat{\theta}, \psi)$                                 ▷ *Update death rate*
3:     $(\mathbf{Y}_t)_{t \in [0,T]} \leftarrow$ SAMPLE-BACKWARD$(\theta, \hat{\theta}, \psi)$          ▷ *Sample from birth process*
4:     **while** reuse paths **do**
5:        $\mathcal{L}_{\mathrm{MM}}(\theta) \leftarrow \mathcal{L}_{\mathrm{MM}}((\mathbf{Y}_t)_{t \in [0,T]}; \theta), \ \mathcal{L}_{\mathrm{TD}}(\theta) \leftarrow \mathcal{L}_{\mathrm{TD}}((\mathbf{Y}_t)_{t \in [0,T]}; \theta)$    ▷ *Compute loss*
6:        Update $\theta$ using $\nabla_\theta(\mathcal{L}_{\mathrm{MM}} + \mathcal{L}_{\mathrm{TD}})$            ▷ *Train forward potential*
7:     $(\mathbf{X}_t)_{t \in [0,T]} \leftarrow$ SAMPLE-FORWARD$(\theta, \hat{\theta}, \psi)$           ▷ *Sample from kill process*
8:     **while** reuse paths **do**
9:        $\mathcal{L}_{\mathrm{MM}}(\hat{\theta}) \leftarrow \mathcal{L}_{\mathrm{MM}}((\mathbf{X}_t)_{t \in [0,T]}; \hat{\theta}), \ \mathcal{L}_{\mathrm{TD}}(\hat{\theta}) \leftarrow \mathcal{L}_{\mathrm{TD}}((\mathbf{X}_t)_{t \in [0,T]}; \hat{\theta})$
10:       Update $\hat{\theta}$ using $\nabla_{\hat{\theta}}(\mathcal{L}_{\mathrm{MM}} + \mathcal{L}_{\mathrm{TD}})$          ▷ *Train backward potential*

---

**Limitations of UDSB-TD.** Although theoretically grounded, UDSB-TD has some limitations. Like in Liu et al. (2022), it needs estimates of $\log \varphi_t^n, \log \hat{\varphi}_t^n$, and not only of $\nabla \log \varphi_t^n, \nabla \log \hat{\varphi}_t^n$, in order to update $\hat{\Psi}$ and $\Psi$. While the TD loss allows for that, it is less stable than the MM loss and, more importantly, requires estimates of $\rho_0, \rho_1$ which might not be available in practice. Furthermore, the formula (11) used to update $\Psi$ might be numerically unstable, especially in high dimensions.

**Heuristic estimation of $\Psi$.** To circumvent these limitations, we introduce another algorithm (UDSB-F) that exhibits a better behavior w.r.t. the dimension of the problem and does not require the estimation of $\log \varphi_t^n, \log \hat{\varphi}_t^n$. While we do not prove the theoretical validity of this new procedure, we verify empirically in Appendix G that its results are consistent with UDSB-TD in small dimensions. Following (8), we know that, at equilibrium, the update on the killing rate is given by $x \mapsto \Psi_t/\varphi_t(x)$. Therefore, we propose to approximate this ratio by $x \mapsto g_{\zeta,t}(x)$, where $g_\zeta$ is learnable. Rewriting (11) at equilibrium, we have

$$\int_0^1 \int_{\mathbb{R}^d} k(x)(\Psi_t/\varphi_t(x)) p_t(x)\mathrm{d}x = \mu_0(\mathbb{R}^d) - \mu_1(\mathbb{R}^d).$$

This suggests considering the loss $\mathcal{L}(\zeta) = \int_0^1 \mathbb{E}[k(\mathbf{X}_t) g_{\zeta,t}(\mathbf{X}_t)]\mathrm{d}t - \mu_0(\mathbb{R}^d) + \mu_1(\mathbb{R}^d)$. We again highlight that minimizing this loss does not ensure that $g_{\zeta,t}$ is the optimal update. However, using this loss we remark that we no longer need to estimate $\log \varphi_t^n, \log \hat{\varphi}_t^n$. Therefore, we can drop the TD loss and make the algorithm more scalable by not requiring estimates of $\rho_0, \rho_1$. The algorithm performing the revised update of the killing rate is presented fully in Appendix F.

## 6 EXPERIMENTS

In this section, we assess the performance of our UDSB solver in two tasks: the reconstruction of simple dynamics in the plane and the modeling of cellular responses to a cancer drug. We defer a thorough comparison of the two schemes discussed in Section 4 (UDSB-TD and UDSB-F) to Appendix G and henceforth utilize UDSB-F algorithm due to its enhanced stability and wider applicability.

**Synthetic dynamics.** We first consider the toy dataset displayed in Fig. 3a, consisting of 3 groups of points that are known to move along the horizontal axis. Standard SBs easily reconstruct the dynamics whenever representatives of all groups appear in observed marginals. However, some clusters may not appear in the empirical distribution. Under-representation could be, for instance, attributable to the exceedingly small number of samples considered or be related to other experimental factors. We

| Methods | Intermediate time (**24h**) | | Final time (**48h**) | |
|---|---|---|---|---|
| | MMD ↓ | $W_\varepsilon$ ↓ | MMD ↓ | $W_\varepsilon$ ↓ |
| Chen et al. (2021) | 3.25e-2 (0.03e-2) | 7.11 (0.02) | 1.87e-2 (0.05e-2) | 6.26 (0.02) |
| Ours (no deaths/births) | 2.45e-2 (0.06e-2) | 6.80 (0.09) | 1.17e-2 (0.03e-2) | 5.66 (0.10) |
| Ours | **2.25e-2** (0.05e-2) | **6.68** (0.08) | **1.13e-2** (0.07e-2) | **5.52** (0.09) |

Table 1: **Cellular dynamics prediction results.** We compare our method against a baseline, a classical SB solver, by measuring the quality of the predicted statuses of cells both at the end of the experiment and at an intermediate time. We use two distributional metrics (MMD, $W_\varepsilon$) and report averages and standard deviations (in parentheses) over 10 runs. Note that, when disabling the death mechanism, our method coincides with Liu et al. (2022).

represent two of these cases: one group does not appear in the end marginal in Fig. 3b, while two are missing in those of Fig. 3e (top-right and middle-left). In both cases, standard SBs fail to capture the correct dynamics, since they generate diagonal trajectories (Fig. 3c, 3f). Orphan particles, which start at the top-left, are in fact forced to reach their closest points in the final marginal, i.e., the middle cluster on the right. This is the product of the assumption placed by SBs that all particles move continuously in the state space. It is not possible to encode the knowledge that a (known) fraction of particles should leave (or enter) the system and that some particles may be extraneous.

Unbalanced SBs provide instead a natural way of identifying regions containing particles that should likely not be considered when learning the diffusion, or ones in which new particles should appear. They consist of death (and birth) priors, which we draw here as *gray* rectangles. A death zone is, for instance, placed close to the orphan marginal in Fig. 3b, while a birth zone stands beside the central cluster on the right of Fig. 3e, since that group of particles does not have a counterpart in the initial marginal. UDSBs then learn trajectories between incomplete marginals, by only using live particles. The statuses of particles are determined by incrementally adjusting the death (and birth) priors to match a predefined amount of mass loss. For example, we consider three different amounts of mass remaining at time $t_1$ (Fig. 3h) and compute UDSB trajectories for each of them (Fig. 3d). In all cases, our algorithm computes correct paths for live particles while ensuring that the number of deaths matches the mass constraint (Fig. 3h).

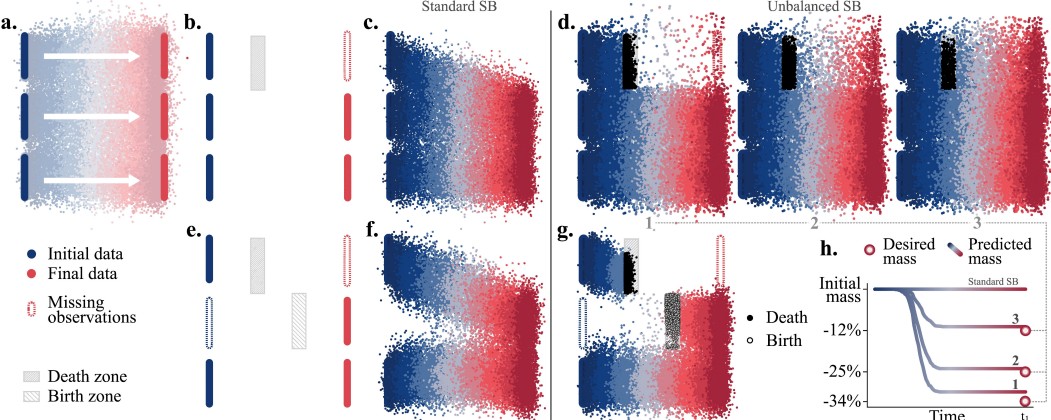

Figure 3: Synthetic datasets. (**a**) Initial (*blue*) and final (*red*) empirical distributions, consisting of 3 groups of points moving horizontally. When some groups are missing in observations (**b**, **e**, dashed), standard SBs fail to learn the correct dynamics (**c**, **f**). By introducing death and birth priors (*gray* rectangles), unbalanced SBs recover, instead, the true evolution law (**d**, **g**). They learn valid interpolations (**d1-3**) between incomplete marginals which also reproduce the (arbitrary) mass loss observed at the end (**h**, circles).

**Cellular dynamics.** Unbalanced SBs have a natural application in the field of cellular biology, where the appearance or disappearance of mass has the physical meaning of cell birth or death. We

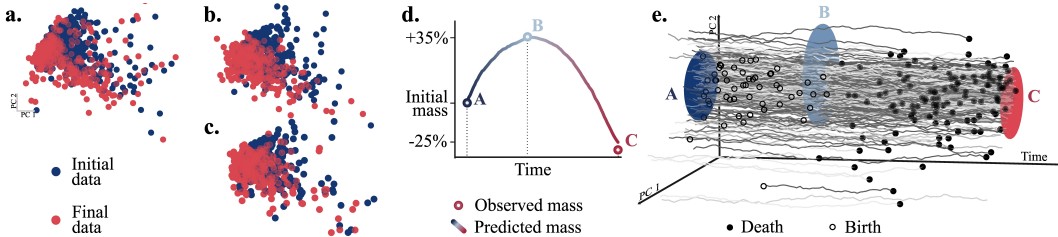

Figure 4: Single-cell response to a cancer drug. (**a**) Cell states (projected along their first two principal components) are measured at the beginning (*blue*) and at the end (*red*). The predictions computed by UDSB-F (**c**) closely match the one of a standard SB solver (**b**) but also account for changes in cell population observed at intermediate times (**d**). The dynamics learned by our unbalanced SB solver (**e**) define the evolution of cell states but also describe death and birth events (*black* and *white* dots). Ellipses identify the $3\sigma$ region around the mean of the observed population at each timepoint.

examine the dataset collected by Lübeck et al. (2022), which tracks the time evolution of single-cell markers of melanoma cells undergoing treatment with several cancer drugs. Cell statuses are recorded at 3 different times and their evolution can be approximated by standard SBs as a continuous diffusion in their (50-dimensional) feature space. However, this model ignores that consecutive measurements capture different cell populations, owing to the death of some cells –caused either by natural or drug-induced reasons– and to the birth of new ones. Trajectories of dying cells are therefore artificially steered to match the final marginal while the presence of newborn cells cannot be properly taken into account. By allowing for jumps in the trajectories of cell statuses, unbalanced SBs overcome, instead, both limitations and more accurately reproduce the measurements. They can, in fact, kill cells with statuses that are dissimilar to the ones found in subsequent measurements while also generating new cells close to observed ones.

We consider a treatment (the drug mix *Trametinib-Erlotinib*) that makes the observed cell population first grow and then shrink over time (dots in Fig. 4d), hence clearly requiring to jointly model cell deaths and births. We plot the temporal evolution of cell trajectories computed by UDSB-F in Fig. 4e. Colored ellipses represent statuses less than 3 standard deviations away from empirical means. Deaths, represented by *black* dots, allow removing some of the cells that are far away from observed statuses, while birth (*white* dots) help introduce new cells located in the densest parts of the state-space. UDSB-F ensures that the number and temporal distribution of deaths/births match observations at both timepoints (Fig. 4d). It learns a standard SB on live particles, which produces better quality predictions of the end marginal (Table 1, 48h) compared to the (balanced) SB solver by Chen et al. (2021). More importantly, the reconstruction quality of the intermediate marginal (Table 1), which is not seen during training, also improves. Interestingly, if we sample trajectories from UDSB-F but disregard deaths and births, the quality of the predictions deteriorates pointing to the role of the coffin state in improving the quality of learned dynamics.

In cases where cell births are not observed or their number is negligible, we can compute UDSBs using the more theoretically-sound version of our algorithm: UDSB-TD. In the appendix, we consider one such case, i.e., treatment with drug *Ulixertinib*, which involves a substantial (>40%) decrease in cell count, and compare the performances of UDSB-TD and UDSB-F.

## 7 Conclusion

In conclusion, this work addresses data transport in the challenging unbalanced setting, essential for accurately tracking population changes in biology applications. Our key contribution extends the IPF procedure for modeling dynamics with time-varying mass information, using time-reversal formulas for diffusions with killing and birth terms. We present efficient algorithms, applied to tasks like modeling cellular responses to cancer drugs and simulating virus spread.

Furthermore, our work establishes connections between the SB formulation and other fields, opening research avenues, including mean-field stochastic games and connections with stochastic optimal control and Wasserstein geometry, deepening the theoretical understanding of SB problems on non-Euclidean spaces.

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

# APPENDIX

In these appendices, we give more details on the stochastic processes we investigate, better describe their links with the Schrödinger bridge problem, and present additional experiments. Notations are defined in Appendix A. In Appendix B, we prove the existence of diffusions with killing under mild assumptions, by using general theoretical results from Ethier and Kurtz (2009). We then prove time-reversal formulas (Appendix C). Discretization schemes of diffusions with killing, as well as the link with their time reversal, are discussed in Appendix D. Appendix E recalls related literature and compares it to this work. Appendix F examines instead the implementation of UDSB-TD and supplies additional details on the more scalable heuristic estimation of $\Psi$, which we use in UDSB-F. Finally, Appendix G includes additional experiments and details datasets, prior processes, models, hyper-parameters, and evaluation metrics.

## A   NOTATION

We denote $C_b^k(\mathbb{R}^d, \mathbb{R})$ the set of functions which are $k$ differentiable and bounded. Similarly, we denote $C_b^k(\mathbb{R}^d, \mathbb{R})$ the set of functions which are $k$ differentiable and compactly supported. The set $C_0^k(\mathbb{R}^d, \mathbb{R})$ denotes the functions which are $k$ differentiable and vanish when $\|x\| \to +\infty$. For any $A \subset \mathbb{R}^d$, we denote $\mathrm{cl}(A)$ its closure. Similarly, we denote $\mathrm{int}(A)$ its interior. Finally, we define $\partial A = \mathrm{cl}(A) \cap \mathrm{int}(A)^c$, the topological boundary of $A$. We denote by $\bar{B}(a\,;r)$ the closed ball with center $a \in \mathbb{R}^d$ and radius $r > 0$. We denote by $B(a\,;r)$ the associated open ball. We denote $\hat{\mathbb{R}}^d = \mathbb{R}^d \cup \{\infty\}$, the one-point compactification of $\mathbb{R}^d$. We refer to Kelley (2017) for details on this construction. We simply note that $f \in C(\hat{\mathbb{R}}^d)$, if $f \in C(\mathbb{R}^d)$ and $f - f(\infty) \in C_0(\mathbb{R}^d)$ and that $f \in C^k(\hat{\mathbb{R}}^d)$ for any $k \in \mathbb{N}$ if the restriction of $f$ to $\mathbb{R}^d$ is in $C^k(\mathbb{R}^d)$ and $f \in C(\hat{\mathbb{R}}^d)$. We recall that in our context, $\{\infty\}$ will play the role of a *cemetery* state. The space of right continuous with left limit functions on $(0, +\infty]$ with valued in $\mathsf{E}$ where $\mathsf{E}$ is a topological space is denoted $\mathrm{D}((0, +\infty], \mathsf{E})$, we refer to Ethier and Kurtz (2009, p.116) for more details on the topology of this space. For a measure $\mu$ on a measurable space $\mathsf{E}$ and $f$ a measurable function on $\mathsf{E}$ such that $\int_\mathsf{E} |f|(x) \mathrm{d}\mu(x) < +\infty$, we denote $\mu(f) = \int_\mathsf{E} f(x) \mathrm{d}\mu(x)$.

Next, we define a *solution of the martingale problem* associated with $\mathcal{A}$, in the sense of Ethier and Kurtz (2009). We only consider the case where the solutions are right continuous. We assume that $\mathsf{E}$ is a metric space and let $\mathcal{A} \subset \mathsf{F}(\mathsf{E}) \times \mathsf{F}(\mathsf{E})$, where $\mathsf{F}(\mathsf{E})$ is the space of real-valued measurable functions on $\mathsf{E}$. A $\mathsf{E}$-valued right continuous stochastic process $(\mathbf{X}_t)_{t \geq 0}$ is a solution of the martingale associated with $\mathcal{A}$ if for any $(f, g) \in \mathcal{A}$, $(f(\mathbf{X}_t) - \int_0^t g(\mathbf{X}_s)\mathrm{d}s)_{t \geq 0}$ is a martingale with respect to its own filtration. In layman terms, in the case where $\mathcal{A}$ is a function, this means that $f(\mathbf{X}_t)$ can be written as

$$f(\mathbf{X}_t) = \int_0^t \mathcal{A}(f)(\mathbf{X}_s)\mathrm{d}s + \mathbf{M}_t,$$

where $(\mathbf{M}_t)_{t \geq 0}$ is a martingale. The notion of *solution of a martingale problem* is associated with the notion of *weak* solution to a Stochastic Differential Equation (SDE). We refer to Stroock and Varadhan (1997) for an extensive study of SDEs from the point of view of martingale problems.

## B   EXISTENCE OF DIFFUSIONS WITH KILLING

In this section, we prove the existence of killed diffusions using the theory of infinitesimal generators. We will leverage general results from (Ethier and Kurtz, 2009). We begin by introducing the *infinitesimal generator* we are going to study. Let $\mathcal{A} : C_b^2(\mathbb{R}^d, \mathbb{R}) \to C_c(\mathbb{R}^d, \mathbb{R})$ be given for any $f \in C_c^2(\mathbb{R}^d, \mathbb{R})$ and $x \in \mathbb{R}^d$ by

$$\mathcal{A}(f)(x) = \tfrac{1}{2}\Delta f(x) + \langle b(x), \nabla f(x) \rangle - k(x)f(x), \tag{12}$$

with $b \in C(\mathbb{R}^d, \mathbb{R}^d)$ called the *drift function* and $k \in C(\mathbb{R}^d, \mathbb{R}_+)$ called the *killing rate*. The regularity assumptions are summarized in the following hypothesis.

**Assumption B.1.** $b \in C(\mathbb{R}^d, \mathbb{R}^d)$ and $k \in C(\mathbb{R}^d, \mathbb{R}_+)$.

Contrary to a classical diffusion associated with a Itô Stochastic Differential Equation (SDE), (12) incorporates a zero order term $x \mapsto -k(x)f(x)$. The fact that $k$ is non-negative is key to the rest of our study. Before stating our main result, we begin by stating a *maximum principle* for $\mathcal{A}$.

**B.1 A Maximum Principle.** In order to apply (Ethier and Kurtz, 2009, Theorem 5.4), we need to ensure that $\mathcal{A}$ satisfies a maximum principle. Such results are ubiquitous in the analysis of Partial Differential Equations (PDEs). In what follows, we use an extension of the Hopf maximum principle, see (Protter and Weinberger, 2012, Theorem 5). For completeness, we provide the proof of this well-known result in our setting following Protter and Weinberger (2012). We start with the following lemma.

**Lemma B.2.** *Assume* Assumption B.1. *Let* $\mathsf{U} \subset \mathbb{R}^d$ *be an open set and* $f \in \mathrm{C}^2(\mathrm{cl}(\mathsf{U}), \mathbb{R})$. *Assume that for any* $x \in \mathsf{U}$, *we have* $\mathcal{A}(f)(x) > 0$. *Assume that there exists* $x_0 \in \mathrm{cl}(\mathsf{U})$ *such that* $f(x_0) = \sup\{f(x) : x \in \mathrm{cl}(\mathsf{U})\}$. *Then* $x_0 \in \partial\mathsf{U}$.

*Proof.* First, assume that $x_0 \in \mathsf{U}$. Then, we have $\nabla f(x_0) = 0$ and $\nabla^2 f(x_0) \prec 0$ and therefore, $\Delta f(x_0) \leq 0$. Since $k \in \mathrm{C}(\mathbb{R}^d, \mathbb{R}_+)$, we get that $\mathcal{A}(f)(x_0) \leq 0$, which is a contradiction. Hence $x_0 \in \partial\mathsf{U}$. $\square$

This lemma is also called the *weak maximum principle*. Equipped with Lemma B.2, we are now ready to prove the main result of this section (Protter and Weinberger, 2012, Theorem 6).

**Proposition B.3.** *Assume* Assumption B.1. *Let* $\mathsf{U} \subset \mathbb{R}^d$ *be an open bounded connected set and* $f \in \mathrm{C}^2(\mathrm{cl}(\mathsf{U}), \mathbb{R})$. *Assume that for any* $x \in \mathsf{U}$, $\mathcal{A}(f)(x) \geq 0$. *Assume that there exists* $x_0 \in \mathsf{U}$ *such that* $f(x_0) = \sup\{f(x) : x \in \mathrm{cl}(\mathsf{U})\}$. *Then* $f$ *is constant on* $\mathsf{U}$.

*Proof.* In this proof, we are going to proceed by contradiction. We construct a function and a domain which contradicts Lemma B.2. First, we start by identifying an open set in $\mathsf{U}$ such that at least one element of the boundary is a maximizer of $f$ and every element in the interior of the open set is *not* a maximizer.

Let $x_0 \in \mathsf{U}$ such that $f(x_0) = M = \sup\{f(x) : x \in \mathrm{cl}(\mathsf{U})\}$ and $x_1 \in \mathsf{U}$ such that $f(x_1) < M$. Since $\mathsf{U}$ is connected, there exists $\gamma \in \mathrm{C}([0,1], \mathsf{U})$ such that $\gamma(0) = x_0$ and $\gamma(1) = x_1$. Since $\gamma([0,1])$ is compact and $\mathsf{U}^c$ is closed, there exists $\delta > 0$ such that for any $t \in [0,1]$ and $x \notin \mathsf{U}$, $\|\gamma(t) - x\| > \delta$. Denote $t_0 = \sup\{f(\gamma(t)) = M : t \in [0,1]\}$. Note that $\{f(\gamma(t)) = M : t \in [0,1]\}$ is not empty since $0 \in \{f(\gamma(t)) = M : t \in [0,1]\}$. Let $t^\star \in (t_0, 1]$ such that $\|\gamma(t^\star) - \gamma(t_0)\| \leq \delta/2$. In what follows, we denote $x^\star = \gamma(t^\star)$.

Next, we define $R \in \mathrm{C}([0, \delta/2], \mathbb{R})$ given for any $s \in [0, \delta/2]$ by $R(s) = \sup\{f(x^\star + sz) : \|z\| \leq 1\}$, we emphasize that $\mathrm{B}(x^\star; \delta/2) \subset \mathsf{U}$ and therefore $R$ is well-defined. Since $\gamma(t_0) \in \mathrm{B}(x^\star; \delta/2)$ and $f(\gamma(t_0)) = M$, we have $R(\delta/2) = M$. Note that $R$ is non-decreasing and denote $s^\star = \inf\{s \in [0, \delta/2] : R(s) = M\}$. By continuity, there exists $x_1 \in \mathsf{U}$ such that $\|x_1 - x^\star\| = s^\star$ and $f(x_1) = M$. In addition, by definition of $s^\star$, for any $x \in \mathrm{B}(x^\star; s^\star)$, $f(x) < M$.

In the rest of the proof, we first present the easier case, where we assume that there exists a *unique* $x_1 \in \mathsf{U}$ such that $\|x_1 - x^\star\| = s^\star$ and $f(x_1) = M$. The general case will require one more construction. Let $r_1 = s^\star/2$ and note that $\bar{\mathrm{B}}(x_1; r_1) \subset \mathsf{U}$. The set $\mathsf{V} = \mathrm{B}(x_1; r_1)$ is the open set on which we are going to apply Lemma B.2. We are now going to define a function $\tilde{f} \in \mathrm{C}^2(\mathrm{cl}(\mathsf{V}), \mathbb{R})$ such that (a) for any $x \in \mathsf{V}$, $\mathcal{A}(\tilde{f})(x) > 0$. (b) $\tilde{f}$ admits a maximizer in $\mathsf{V}$. This will contradict Lemma B.2. For any $\alpha > 0$, we introduce the following auxiliary function $w_\alpha \in \mathrm{C}^2(\mathrm{B}(x^\star; \delta), \mathbb{R})$ given for any $x \in \mathrm{B}(x^\star; \delta)$ by

$$w_\alpha(x) = \exp[-\alpha\|x - x^\star\|^2/2] - \exp[-\alpha(s^\star)^2/2].$$

Since $\mathsf{V} \subset \mathrm{B}(x^\star; \delta)$ we have that $w_\alpha$ is well-defined on $\mathsf{V}$. In addition, using this result and the fact that for any $x \in \mathsf{V}$, $\|x - x^\star\| \geq \|x_1 - x^\star\| - \|x_1 - x\| \geq s^\star/2$ we have for any $x \in \mathsf{V}$ and $\alpha > 0$

$$\mathcal{A}(w_\alpha)(x) \geq \exp[-\alpha\|x - x^\star\|^2/2](\alpha^2\|x - x^\star\|^2 + \alpha(d - C\|x - x^\star\|) - C)$$

$$\geq \exp[-\alpha\|x - x^\star\|^2/2](\alpha^2(s^\star)^2/4 + \alpha(d - C\delta/2) - C).$$

where $C = \sup\{\|b(x)\| + k(x) : x \in \mathrm{cl}(\mathsf{U})\}$. Hence, there exists $\alpha_0 > 0$ such that for any $x \in \mathsf{V}$, $\mathcal{A}(w_{\alpha_0})(x) > 0$. In addition, for any $x \in \partial\mathsf{V}$, if $\|x - x^\star\| \leq s^\star$, we have $w_{\alpha_0}(x) \leq 1 - \exp[-\alpha(s^\star)^2/2]$. If $\|x - x^\star\| > s^\star$, then $w_{\alpha_0}(x) < 0$.

We let $\xi = \inf\{f(x) \, : \, x \in \partial V \cap \bar{B}(x^\star\,;s^\star)\}$. If $x \in B(x^\star\,;s^\star)$ then $f(x) < M$. Similarly, because we have assume that there exists a unique $x_1 \in U$ such that $\|x_1 - x^\star\| = s^\star$ and $f(x_1) = M$ and $x_1 \notin \partial V$, we have that if $x \in \partial B(x^\star\,;s^\star) \cap \partial V$, $f(x) < M$. Hence, $\xi < M$. Let $\varepsilon > 0$ such that $\xi + \varepsilon(1 - \exp[-\alpha(s^\star)^2/2]) < M$ and we get that for any $x \in \bar{B}(x^\star\,;s^\star) \cap \partial V$, $f(x) + \varepsilon w_{\alpha_0}(x) < M$. In addition, for any $x \in \partial V \cap \bar{B}(x^\star\,;s^\star)^c$, we have $f(x) + \varepsilon w_{\alpha_0}(x) \leq M - \varepsilon w_{\alpha_0}(x) < M$. Therefore, we get that for any $x \in \partial V$, $f(x) + \varepsilon w_{\alpha_0}(x) < M$. In addition, $f(x_1) + \varepsilon w_{\alpha_0}(x_1) = M$, since $w_{\alpha_0}(x_1) = 0$. In addition, we have that for any $x \in V$, $\mathcal{A}(f + \varepsilon w_{\alpha_0})(x) > 0$. We have a contradiction with Lemma B.2 upon letting $\tilde{f} = f + \varepsilon w_{\alpha_0}$. As emphasize above, we cannot conclude since we made the additional assumption that $x_1$ was the *unique* element such that $\|x_1 - x^\star\| = s^\star$ and $f(x_1) = M$.

In the rest of the proof, we treat the general case which requires one more construction. Let $\tilde{s} = s^\star/2$ and consider $\tilde{x} \in [x^\star, x_1]$ such that $\|\tilde{x} - x_1\| = s^\star/2$. The ball $B(\tilde{x}\,;\tilde{s})$ is going to replace $B(x^\star\,;s^\star)$ in the proof above. Next, we consider $r_1 = \tilde{s}/2$ and note that $\bar{B}(x_1\,;r_1) \subset U$. The set $V = B(x_1\,;r_1)$ is the open set on which we are going to apply Lemma B.2. We are now going to define a function $\tilde{f} \in C^2(\mathrm{cl}(V), \mathbb{R})$ such that (a) for any $x \in V$, $\mathcal{A}(\tilde{f})(x) > 0$. (b) $\tilde{f}$ admits a maximizer in $V$. This will contradict Lemma B.2. For any $\alpha > 0$, we introduce the following auxiliary function $w_\alpha \in C^2(B(\tilde{x}\,;\delta), \mathbb{R})$ given for any $x \in B(\tilde{x}\,;\delta)$ by

$$w_\alpha(x) = \exp[-\alpha\|x - \tilde{x}\|^2/2] - \exp[-\alpha\tilde{s}^2/2],$$

Since $V \subset B(\tilde{x}\,;\delta)$ we have that $w_\alpha$ is well-defined on $V$. In addition, using this result and the fact that for any $x \in V$, $\|x - \tilde{x}\| \geq \|x_1 - \tilde{x}\| - \|x_1 - x\| \geq \tilde{s}/2$ we have for any $x \in V$ and $\alpha > 0$

$$\mathcal{A}(w_\alpha)(x) \geq \exp[-\alpha\|x - \tilde{x}\|^2/2](\alpha^2\|x - \tilde{x}\|^2 + \alpha(d - C\|x - \tilde{x}\|) - C)$$
$$\geq \exp[-\alpha\|x - \tilde{x}\|^2/2](\alpha^2(\tilde{s})^2/4 + \alpha(d - C\delta/2) - C).$$

where $C = \sup\{\|b(x)\| + k(x) \, : \, x \in \mathrm{cl}(U)\}$. Hence, there exists $\alpha_0 > 0$ such that for any $x \in V$, $\mathcal{A}(w_{\alpha_0})(x) > 0$. In addition, for any $x \in \partial V$, if $\|x - \tilde{x}\| \leq \tilde{s}$, we have $w_{\alpha_0}(x) \leq 1 - \exp[-\alpha(\tilde{s})^2/2]$. If $\|x - \tilde{x}\| > \tilde{s}$, then $w_{\alpha_0}(x) < 0$.

We let $\xi = \inf\{f(x) \, : \, x \in \partial V \cap \bar{B}(\tilde{x}\,;\tilde{s})\}$. If $x \in \bar{B}(\tilde{x}\,;\tilde{s})$ and $x \neq x_1$ then $f(x) < M$. Hence, $\xi < M$. Let $\varepsilon > 0$ such that $\xi + \varepsilon(1 - \exp[-\alpha(\tilde{s})^2/2]) < M$ and we get that for any $x \in \bar{B}(\tilde{x}\,;\tilde{s}) \cap \partial V$, $f(x) + \varepsilon w_{\alpha_0}(x) < M$. In addition, for any $x \in \partial V \cap \bar{B}(\tilde{x}\,;\tilde{s})^c$, we have $f(x) + \varepsilon w_{\alpha_0}(x) \leq M - \varepsilon w_{\alpha_0}(x) < M$. Therefore, we get that for any $x \in \partial V$, $f(x) + \varepsilon w_{\alpha_0}(x) < M$. In addition, $f(x_1) + \varepsilon w_{\alpha_0}(x_1) = M$, since $w_{\alpha_0}(x_1) = 0$. In addition, we have that for any $x \in V$, $\mathcal{A}(f + \varepsilon w_{\alpha_0})(x) > 0$. We have a contradiction with Lemma B.2 upon letting $\tilde{f} = f + \varepsilon w_{\alpha_0}$, which concludes the proof. $\square$

In Proposition B.3, we prove a maximum principle for the *elliptic* infinitesimal generator (12). In order to define the backward Kolmogorov equation associated with the process, we need to consider the *parabolic* infinitesimal generator instead, given for any $f \in C_c^2((0, +\infty) \times \mathbb{R}^d, \mathbb{R})$ and $t > 0$, $x \in \mathbb{R}^d$ by

$$\mathcal{A}(f)(x) = \partial_t f(t, x) + \tfrac{1}{2}\Delta f(t, x) + \langle b(t, x), \nabla f(t, x)\rangle - k(x)f(t, x). \tag{13}$$

It can be shown that the generator (13) also satisfies a strong version of the maximum principle (Nirenberg, 1953).

**B.2 Infinitesimal Generator.** Using Proposition B.3, we are ready to define diffusion processes with killing. The notion of *solution to a martingale problem* are recalled in Appendix A. We also denote $\hat{\mathcal{A}}$, the extension of $\mathcal{A}$ given in (12) to the one point compactification $\hat{\mathbb{R}}^d$ and defined as follows: for any $f \in C(\hat{\mathbb{R}}^d)$ and $x \in \mathbb{R}^d$, we have

$$\hat{\mathcal{A}}(f)(x) = \mathcal{A}(f - f(\infty))(x), \qquad \hat{\mathcal{A}}(f)(\infty) = 0. \tag{14}$$

The first equation can be rewritten for any $f \in C(\hat{\mathbb{R}}^d)$ and $x \in \mathbb{R}^d$

$$\mathcal{A}(f)(x) = \tfrac{1}{2}\Delta f(x) + \langle b(x), \nabla f(x)\rangle - k(x)(f(x) - f(\infty)). \tag{15}$$

We have the following theorem.

**Theorem B.4.** *Assume* Assumption B.1. *For any probability measure $\nu$ on $\hat{\mathbb{R}}^d$ exists $(\mathbf{X}_t)_{t \geq 0} \in$ $\mathrm{D}((0, +\infty], \hat{\mathbb{R}}^d)$ solution to the martingale problem associated with $\hat{\mathcal{A}}$ with initial condition $\nu$.*

*Proof.* The space $\mathbb{R}^d$ is locally compact and separable. The generator $\mathcal{A}$ is a linear operator and $\mathrm{C}_c^2(\mathbb{R}^d, \mathbb{R})$ is dense in $\mathrm{C}_c(\mathbb{R}^d, \mathbb{R})$. In addition, using Proposition B.3, we have that $\mathcal{A}$ satisfies the positive maximum principle. We conclude the proof upon applying (Ethier and Kurtz, 2009, Theorem 5.4, p.199). □

Using (Ethier and Kurtz, 2009, Section 4, p.182) it is also possible to show the uniqueness (in some sense) of the process under mild assumptions.

## C  TIME REVERSAL OF DIFFUSIONS WITH KILLING

Now that we have established the existence of a killed diffusion in Appendix B, we study its time-reversal. The main goal of this section is to identify its associated infinitesimal generator. Before stating these results, we provide some results on the *extended* generator (15). In particular, we derive the associated forward and backward Kolmogorov equations.

**C.1  Feynman-Kac Semigroups.**    We start by giving a useful representation of killed diffusions using Feynman-Kac semigroups. We will consider the following assumptions.

**Assumption C.1.**  $b \in \mathrm{C}(\mathbb{R}^d, \mathbb{R})$, $k$ is bounded and there exists $\mathsf{L} \geq 0$ such that for any $x, y \in \mathbb{R}^d$, $\|b(x) - b(y)\| \leq \mathsf{L} \|x - y\|$.

**Assumption C.2.**  For every $f \in \mathrm{C}(\mathbb{R}^d, \mathbb{R})$ and $T > 0$, there exists $v \in \mathrm{C}([0, T] \times \mathbb{R}^d, \mathbb{R})$ and $v \in \mathrm{C}^{1,2}((0, T] \times \mathbb{R}^d, \mathbb{R})$ such that for any $t \in (0, T]$ and $x \in \mathbb{R}^d$, we have $\mathcal{A}(v)(t, x) = 0$ with $\mathcal{A}$ given by (13) and $v(0, x) = f(x)$. In addition, there exist $C \geq 0$ and $\mathtt{m} \in [0, 1/(2Td)]$ such that for any $x \in \mathbb{R}^d$, $\sup\{\|v(t, x)\| : t \in [0, T]\} \leq C \exp[\mathtt{m}\|x\|^2]$.

Note that conditions on the existence and regularity of solutions to the parabolic equation $\mathcal{A}(v) = 0$ and $\mathcal{A}$ given by (13) have been established in the literature under various assumptions on the coefficients of $\mathcal{A}$, see (Friedman, 2012) or (Dynkin and Dynkin, 1965, Theorem 13.16). The following theorem is called the *Feynman-Kac* representation theorem and draws an explicit link between SDEs and PDEs. It can be seen as a generalization of the backward Kolmogorov equation, see (Karatzas et al., 1991, Theorem 7.6).

**Proposition C.3** (Feynman-Kac formula). *Assume* Assumption B.1, Assumption C.1 *and* Assumption C.2. *For any $f \in \mathrm{C}(\mathbb{R}^d, \mathbb{R})$ and $T > 0$, there exists a unique solution $v \in \mathrm{C}([0, T] \times \mathbb{R}^d, \mathbb{R})$ and $v \in \mathrm{C}^{1,2}((0, T] \times \mathbb{R}^d, \mathbb{R})$ to $\mathcal{A}(v) = 0$. In addition, we have that for any $t \in [0, T]$ and $x \in \mathbb{R}^d$*

$$v(t, x) = \mathbb{E}[f(\mathbf{X}_T) \exp[-\int_t^T k(\mathbf{X}_s^0) \mathrm{d}s]], \qquad (16)$$

*where $(\mathbf{X}_t)_{t \geq 0}$ is the unique (strong) solution to $\mathrm{d}\mathbf{X}_s^0 = b(\mathbf{X}_s^0)\mathrm{d}s + \mathrm{d}\mathbf{B}_s$, $\mathbf{X}_t^0 = x$ and $(\mathbf{B}_s)_{s \geq 0}$ is a $d$-dimensional Brownian motion.*

The Feynman-Kac formula in Proposition C.3 will be used to establish a backward Kolmogorov equation for *killed* diffusions. We conclude this section by giving an explicit form of the semigroup of killed diffusions under assumptions on the killing rate. More precisely, we are going to show that these diffusions can be seen as a *reweighting* of the original unconstrained diffusions. To do so, we need to precise the notion of semigroup. We start with the notion of Markov kernel.

**Definition C.4.**  Let $\mathsf{E}, \mathsf{F}$ be metric spaces. $\mathrm{K} : \mathsf{E} \times \mathcal{B}(\mathsf{F}) \to [0, 1]$ is called a Markov kernel if for any $x \in \mathsf{E}$, $\mathrm{K}(x, \cdot)$ is a probability measure and for any $\mathsf{A} \in \mathcal{B}(\mathsf{F})$ we have that $\mathrm{K}(\cdot, \mathsf{A})$ is measurable.

We are now ready to define the notion of semigroup. We refer to (Ethier and Kurtz, 2009, Chapter 4) for a discussion.

**Definition C.5.**  Let $(\mathrm{P}_t)_{t \geq 0}$ be a collection of Markov kernels with $\mathsf{E} = \mathsf{F}$ such that for any $x \in \mathsf{E}$, $\mathrm{P}_0(x, \cdot) = \delta_x$ and for any $s, t \geq 0$, $x \in \mathsf{E}$ and $\mathsf{A} \in \mathcal{B}(\mathsf{E})$ we have

$$\mathrm{P}_{t+s}(x, \mathsf{A}) = \int_{\mathsf{E}} \mathrm{P}_t(y, \mathsf{A})\mathrm{P}_s(x, \mathrm{d}y).$$

A semigroup on $\mathsf{E}$ is said to be strongly continuous on $\mathrm{C}_0(\mathsf{E})$ if for any $f \in \mathrm{C}_0(\mathsf{E})$, $\lim_{t \to 0} \mathrm{P}_t(f) = f$ uniformly and $\mathrm{P}_t(f) \in \mathrm{C}_0(\mathsf{E})$.

The notion of semigroup is intrinsically linked with the one of infinitesimal generator. In particular, we define the generator associated with a semigroup $(\mathrm{P}_t)_{t \geq 0}$ as

$$\mathcal{A}(f) = \lim_{t \to 0} (\mathrm{P}_t(f) - f)/t.$$

The *domain* of $\mathcal{A}$, denoted $\mathcal{D}(\mathcal{A})$, is the space of functions for which the limit is well-defined. This definition of the generator justifies the notation "$\mathrm{P}_t = \mathrm{e}^{t\mathcal{A}}$".

In Theorem B.4, we have shown the existence of (Markov) killed diffusions based on mild assumptions on the generator. The next result goes further and gives an *explicit* representation of the associated semigroup. The following proposition is adapted from (Sznitman, 1998, Theorem 1.1).

**Proposition C.6.** *Assume* Assumption B.1 *and* Assumption C.1. *Let* $(\mathbf{X}_t^0)_{t \geq 0}$ *be the unique (strong) solution to* $\mathrm{d}\mathbf{X}_t^0 = b(\mathbf{X}_t^0)\mathrm{d}t + \mathrm{d}\mathbf{B}_t$ *and define* $(\mathrm{P}_t)_{t \geq 0}$ *such that for any* $t \geq 0$ *and* $f \in \mathrm{C}_0(\mathbb{R}^d)$

$$\mathrm{P}_t(f) = \mathbb{E}[f(\mathbf{X}_t^0) \exp[-\int_0^t k(\mathbf{X}_s^0)\mathrm{d}s]]. \tag{17}$$

*Then* $(\mathrm{P}_t)_{t \geq 0}$ *can be extended into a strongly continuous semigroup on* $\hat{\mathbb{R}}^d$. *We have that* $(\mathrm{P}_t)_{t \geq 0}$ *admits a generator* $\hat{\mathcal{A}}$ *with* $\mathrm{C}_c^2(\mathbb{R}^d) \subset \mathcal{D}(\hat{\mathcal{A}})$ *and* $\hat{\mathcal{A}}$ *given by* (14). *In addition, we have the following* ***perturbation formulas*** *for any* $t \geq 0$, $x \in \mathbb{R}^d$ *and* $f \in \mathrm{C}_0(\mathbb{R}^d)$:

$$\mathrm{P}_t(x, f) = \mathrm{P}_t^0(x, f) - \int_0^t \mathrm{P}_s^0(x, k\mathrm{P}_{t-s}(f))\mathrm{d}s, \tag{18}$$
$$= \mathrm{P}_t^0(x, f) - \int_0^t \mathrm{P}_s(x, k\mathrm{P}_{t-s}^0(f))\mathrm{d}s,$$

*where* $(\mathrm{P}_t^0)_{t \geq 0}$ *is the semigroup associated with* $(\mathbf{X}_t^0)_{t \geq 0}$. *Finally, there exists a Markov process* $(\mathbf{X}_t)_{t \geq 0}$ *on* $\hat{\mathbb{R}}^d$ *associated with the* $\hat{\mathbb{R}}^d$ *extension of* $(\mathrm{P}_t)_{t \geq 0}$.

*Proof.* We first begin by proving (18). We have for any $t \geq 0$

$$\exp[-\int_0^t k(\mathbf{X}_s^0)\mathrm{d}s] = 1 - \int_0^t k(\mathbf{X}_s^0) \exp[-\int_s^t k(\mathbf{X}_u^0)\mathrm{d}u]\mathrm{d}s$$
$$= 1 - \int_0^t k(\mathbf{X}_s^0) \exp[-\int_0^s k(\mathbf{X}_u^0)\mathrm{d}u]\mathrm{d}s.$$

Hence, for any $t \geq 0$, $x \in \mathbb{R}^d$ and $f \in \mathrm{C}_0(\mathbb{R}^d)$, we have

$$\mathrm{P}_t(x, f) = \mathbb{E}[f(\mathbf{X}_t^0)] - \int_0^t \mathbb{E}[f(\mathbf{X}_t^0)k(\mathbf{X}_s^0) \exp[-\int_s^t k(\mathbf{X}_u^0)\mathrm{d}u]]\mathrm{d}s$$
$$= \mathbb{E}[f(\mathbf{X}_t^0)] - \int_0^t \mathbb{E}[k(\mathbf{X}_s^0)\mathbb{E}[f(\tilde{\mathbf{X}}_{t-s}^0) \exp[-\int_0^{t-s} k(\tilde{\mathbf{X}}_u^0)\mathrm{d}u] \mid \tilde{\mathbf{X}}_0^0 = \mathbf{X}_s^0]]\mathrm{d}s$$
$$= \mathbb{E}[f(\mathbf{X}_t^0)] - \int_0^t \mathbb{E}[k(\mathbf{X}_s^0)\mathrm{P}_{t-s}(\mathbf{X}_s^0, f)]\mathrm{d}s,$$

which concludes the proof of (18). Since $k \in \mathrm{C}_b(\mathbb{R}^d, \mathbb{R})$, we have for any $f \in \mathrm{C}_0(\mathbb{R}^d)$ and $x \in \mathbb{R}^d$, $(s, t) \in \mathrm{P}_s(x, k\mathrm{P}_{t-s}^0(f))$ which is continuous and therefore we get that for any $f \in \mathrm{C}_0(\mathbb{R}^d)$ and $x \in \mathbb{R}^d$

$$\lim_{t \to 0} \tfrac{1}{t} \int_0^t \mathrm{P}_s^0(x, k\mathrm{P}_{t-s}(f)) = k(x)f(x). \tag{19}$$

We extend $(\mathrm{P}_t)_{t \geq 0}$ to $\hat{\mathbb{R}}^d$ by denoting for any $f \in \mathrm{C}(\hat{\mathbb{R}}^d), x \in \mathbb{R}^d$ and $t \geq 0$

$$\hat{\mathrm{P}}_t(x, f) = \mathrm{P}_t(x, f) + f(\infty)(1 - \exp[-\int_0^t k(\mathbf{X}_s^0)\mathrm{d}s]).$$

In addition, we let $\mathrm{P}_t(\infty, f) = f(\infty)$. Note that $(\hat{\mathrm{P}}_t)_{t \geq 0}$ defines a strongly continuous semigroup on $\hat{\mathbb{R}}^d$. Using (19) and the fact that $\mathrm{C}_c^2(\mathbb{R}^d) \subset \mathcal{D}(\mathcal{A}^0)$, where $\mathcal{A}^0$ is the generator associated with $(\mathbf{X}_t^0)_{t \geq 0}$, we get that the generator of $(\hat{\mathrm{P}}_t)_{t \geq 0}$ is given for any $f \in \mathrm{C}_c^2(\mathbb{R}^d)$ and $x \in \mathbb{R}^d$ by

$$\mathcal{A}(f)(x) = \mathcal{A}^0(f)(x) - k(x)f(x).$$

We conclude the proof upon using (Ethier and Kurtz, 2009, Theorem 1.1, p.157). □

For an introduction to killed diffusions we refer to (Oksendal, 2013), see also (Karlin and Taylor, 1981; Blumenthal and Getoor, 2007).

Let us comment on the form of (17) in Proposition C.6. Formally speaking, this formula indicates that in order to integrate a function $f$ with respect to the killed diffusion we can consider an integration

w.r.t. to the unconstrained diffusion $(\mathbf{X}_t^0)_{t \geq 0}$ with a *loss of mass* given by the exponential term $\exp[-\int_0^t k(\mathbf{X}_s^0)\mathrm{d}s]$. Consider the limit case where $k = +\infty$ on some domain $\mathsf{A}$ and $k = 0$ otherwise. Then (17) becomes $\mathrm{P}_t(f) = \mathbb{E}[f(\mathbf{X}_t^0)\mathbf{1}_{t < \tau_\mathsf{A}}]$, where $\tau_\mathsf{A} = \inf\{s \geq 0 \,:\, \mathbf{X}_s \in \mathsf{A}\}$.

Equipped with the notion of Feynman-Kac semigroup, we can rewrite (16) in Proposition C.3 as $v(t, x) = \mathrm{P}_{T-t}(x, f)$. This remark is at the basis of the backward Kolmogorov formula for killed diffusions.

**C.2 Forward and Backward Kolmogorov Equations.** In order to derive the time-reversal of a stochastic process it is useful to define its time-reversal (Haussmann and Pardoux, 1986). In what follows, we denote $(\mathbf{X}_t)_{t \geq 0}$, the killed diffusion, i.e., the process associated with $(\mathrm{P}_t)_{t \geq 0}$.

**Proposition C.7.** *Assume* Assumption B.1, Assumption C.1 *and* Assumption C.2. *Let* $(\mathbf{X}_t)_{t \geq 0}$ *be given by* Proposition C.6. *Then, for any* $f \in \mathrm{C}_0(\mathbb{R}^d)$, $T, t \geq 0$ *with* $T \geq t$ *and* $x \in \mathbb{R}^d$

$$\partial_t \mathrm{P}_t(x, f) = \mathcal{A}(\mathrm{P}_t(\cdot, f))(x), \qquad \partial_t \mathrm{P}_{T-t}(x, f) = -\mathcal{A}(\mathrm{P}_{T-t}(\cdot, f))(x), \qquad (20)$$

*with* $\mathcal{A}$ *given by* (15).

The first part of (20) is referred to as the *forward Kolmogorov equation* while the second part is referred to as the *backward Kolmogorov equation*. The proof is a direct consequence of Proposition C.6. Note that under Assumption C.2 we have additional regularity assumptions on $(t, x) \mapsto \mathrm{P}_{T-t}(x, f)$ and therefore its evolution can be made explicit using the expression of $\mathcal{A}$.

**C.3 Reversal of the Fokker-Planck Equation.** In this paragraph, we provide a heuristic derivation of the time-reversal of the killed diffusion process. Under mild assumptions, we get that $\mathrm{P}_t$ admits a density w.r.t the Lebesgue measure denoted $p_t$ for any $t > 0$. Note that $\int_{\mathbb{R}^d} p_t(x)\mathrm{d}x \leq 1$. In addition, we have that

$$\partial_t p_t(x) = -\mathrm{div}(b(t, \cdot)p_t)(x) + \tfrac{1}{2}\Delta p_t(x) - k(x)p_t(x).$$

Note that the right-hand side of the previous equation can formally be identified with the dual operator $\mathcal{A}^\star$. Now, considering the time-reversal of the previous equation, we get

$$\partial_t p_{T-t}(x) = -\mathrm{div}(\{-b(t, \cdot) + \nabla \log p_{T-t}\}p_{T-t})(x) + \tfrac{1}{2}\Delta p_{T-t}(x) + k(x)p_{T-t}(x).$$

Note that the *non-negative* term $k(x)p_t(x)$ is turned into a *non-positive* term $-k(x)p_{T-t}(x)$. This suggests that the *death* behavior of the forward process is turned into a *birth* behavior for the backward process. We will make this statement precise in the next paragraph. This is in contrast with common results which state that if the forward process is in a certain class then so is the backward process. For instance, in the case of a reflected forward process, the backward is also reflected (Lou and Ermon, 2023; Fishman et al., 2023).

**C.4 Time Reversal of Diffusion With Births.** In the previous paragraph, we provided the time-reversal of the *dual* generator in order to derive the backward evolution of the density $p_t$. In this section, we follow the approach of Haussmann and Pardoux (1986) to derive a *pathwise* time-reversal. Doing so we will directly obtain the time-reversal of the generator (and not its dual). We start by recalling the expression of the extended infinitesimal generator, (14). For any $f \in \mathrm{C}^2(\hat{\mathbb{R}}^d)$ and $x \in \hat{\mathbb{R}}^d$ we have

$$\hat{\mathcal{A}}(f)(x) = [\langle b(x), \nabla f(x)\rangle + \tfrac{1}{2}\Delta f(x) - k(x)(f(x) - f(\infty))]\mathbf{1}_{\mathbb{R}^d}(x).$$

In what follows, we let $(\mathbf{X}_t)_{t \geq 0}$ given by Proposition C.6. We denote $S_t = \mathbb{P}[\mathbf{X}_t \in \mathbb{R}^d]$ for any $t \geq 0$. We will also make the following assumption.

**Assumption C.8.** For any $t \geq 0$, the measure $\mu_t$ on $\mathbb{R}^d$ given for any $f \in \mathrm{C}_c(\mathbb{R}^d)$ by $\mu_t[f] = \mathbb{E}[f(\mathbf{X}_t)\mathbf{1}_{\mathbb{R}^d}(\mathbf{X}_t)]$ admits a density $p_t$ w.r.t. the Lebesgue measure. In addition $(t, x) \mapsto p_t(x) \in \mathrm{C}_b^\infty((0, +\infty) \times \mathbb{R}^d, (0, +\infty))$. Finally, we assume that $S_t < 1$ for any $t > 0$.

The following lemma is central to establish our result.

**Lemma C.9.** *Assume* Assumption B.1, Assumption C.1 *and* Assumption C.8. *Then for any* $t > 0$, $h : \mathbb{R}^d \times \{\infty\} \to \mathbb{R}$ *and* $g : \{\infty\} \to \mathbb{R}$ *measurable and bounded, we have*

$$\mathbb{E}[\mathbf{1}_{\mathbb{R}^d}(\mathbf{X}_t)h(\mathbf{X}_t, \infty)g(\infty)] = \mathbb{E}[\mathbf{1}_\infty(\mathbf{X}_t) \textstyle\int_{\mathbb{R}^d} h(x, \mathbf{X}_t)p_t(x)\mathrm{d}x/(1 - S_t)g(\mathbf{X}_t)].$$

*Proof.* Let $t > 0$, $h : \mathbb{R}^d \times \{\infty\} \to \mathbb{R}$ and $g : \{\infty\} \to \mathbb{R}$ measurable and bounded. First, we have

$$\mathbb{E}[\mathbf{1}_{\mathbb{R}^d}(\mathbf{X}_t)h(\mathbf{X}_t, \infty)g(\infty)] = \int_{\mathbb{R}^d} h(x, \infty)g(\infty)p_t(x)\mathrm{d}x$$
$$= \mathbb{E}[\mathbf{1}_{\infty}(\mathbf{X}_t) \int_{\mathbb{R}^d} h(x, \mathbf{X}_t)g(\mathbf{X}_t)p_t(x)\mathrm{d}x]/(1 - S_t),$$

which concludes the proof. $\qquad\square$

Based on this lemma, we are now ready to state the our time-reversal result, extending the approach of Haussmann and Pardoux (1986). We make the following assumption, which ensures that integration by part is valid in our setting.

**Assumption C.10.** For any $f, g \in \mathrm{C}^2(\hat{\mathbb{R}}^d)$ we have

$$\mathbb{E}[\mathbf{1}_{\mathbb{R}^d}(\mathbf{X}_t)\langle \nabla f(\mathbf{X}_t), \nabla g(\mathbf{X}_t)\rangle] = -\mathbb{E}[\mathbf{1}_{\mathbb{R}^d}(\mathbf{X}_t)g(\mathbf{X}_t)(\Delta f(\mathbf{X}_t) + \langle \nabla \log p_t(\mathbf{X}_t), \nabla f(\mathbf{X}_t)\rangle)].$$

We emphasize that Assumption C.10 finds sufficient condition on the parameters of the unconstrained diffusion such that this integration by part formula is true in the unconstrained setting. We leave the extension of such results to the killed case for future work. We are now ready to state our time-reversal formula.

**Proposition C.11.** *Assume* Assumption B.1, Assumption C.1, Assumption C.8 *and* Assumption C.10. *Let* $(\mathbf{X}_t)_{t \geq 0}$ *given by* Proposition C.6, $T > 0$ *and consider* $(\mathbf{Y}_t)_{t \in [0,T]} = (\mathbf{X}_{T-t})_{t \in [0,T]}$. *Then* $(\mathbf{Y}_t)_{t \in [0,T]}$ *is solution to the martingale problem associated with* $\hat{\mathcal{R}}$, *where for any* $f \in \mathrm{C}^2(\hat{\mathbb{R}}^d)$, $t \in (0, T)$ *and* $x \in \hat{\mathbb{R}}^d$ *we have*

$$\mathcal{R}(f)(t, x) = [\langle -b(x) + \nabla \log p_{T-t}(x), \nabla f(x)\rangle + \tfrac{1}{2}\Delta f(x)]\mathbf{1}_{\mathbb{R}^d}(x)$$
$$+ \int_{\mathbb{R}^d} p_{T-t}(\tilde{x})k(\tilde{x})(f(\tilde{x}) - f(\infty))\mathrm{d}\tilde{x}/(1 - S_{T-t})\mathbf{1}_{\infty}(x). \quad (21)$$

*Proof.* Let $f, g \in \mathrm{C}^2(\hat{\mathbb{R}}^d)$. We are going to show that for any $s, t \in [0, T]$ with $t \geq s$

$$\mathbb{E}[(f(\mathbf{Y}_t) - f(\mathbf{Y}_s))g(\mathbf{Y}_s)] = \mathbb{E}[g(\mathbf{Y}_s) \int_s^t \mathcal{R}(f)(u, \mathbf{Y}_u)\mathrm{d}u].$$

This is equivalent to show that for any $s, t \in [0, T]$ with $t \geq s$

$$\mathbb{E}[(f(\mathbf{X}_t) - f(\mathbf{X}_s))g(\mathbf{X}_t)] = \mathbb{E}[-g(\mathbf{X}_t) \int_s^t \mathcal{R}(f)(u, \mathbf{X}_u)\mathrm{d}u].$$

Let $s, t \in [0, T]$, with $t \geq s$. In what follows, we denote for any $u \in [0, t]$ and $x \in \hat{\mathbb{R}}^d$, $g(u, x) = \mathbb{E}[g(\mathbf{X}_t) \mid \mathbf{X}_u = x]$. Using Proposition C.7, we have that for any $u \in [0, t]$ and $x \in \hat{\mathbb{R}}^d$, $\partial_u g(u, x) + \hat{\mathcal{A}}(g)(u, x) = 0$, i.e. $g$ satisfies the backward Kolmogorov equation. For any $u \in [0, t]$ and $x \in \hat{\mathbb{R}}^d$, we have

$$\hat{\mathcal{A}}(fg)(u, x) = \partial_u g(u, x)f(x) + (\langle b(x), \nabla g(u, x)\rangle + \tfrac{1}{2}\Delta g(u, x))f(x)\mathbf{1}_{\mathbb{R}^d}(x)$$
$$+ (\langle b(x), \nabla f(x)\rangle + \tfrac{1}{2}\Delta f(x))g(u, x)\mathbf{1}_{\mathbb{R}^d}(x) + \mathbf{1}_{\mathbb{R}^d}(x)\langle \nabla f(x), \nabla g(u, x)\rangle$$
$$+ \mathbf{1}_{\mathbb{R}^d}(f(x)g(u, x) - f(\infty)g(u, \infty))$$
$$= \partial_u g(u, x)f(x) + \hat{\mathcal{A}}(g)(u, x)f(x) + \mathbf{1}_{\mathbb{R}^d}(x)(f(x) - f(\infty))g(u, \infty)$$
$$+ (\langle b(x), \nabla f(x)\rangle + \tfrac{1}{2}\Delta f(x))g(u, x)\mathbf{1}_{\mathbb{R}^d}(x) + \mathbf{1}_{\mathbb{R}^d}(x)\langle \nabla f(x), \nabla g(u, x)\rangle$$
$$= (\langle b(x), \nabla f(x)\rangle + \tfrac{1}{2}\Delta f(x))g(u, x)\mathbf{1}_{\mathbb{R}^d}(x) + \mathbf{1}_{\mathbb{R}^d}(x)\langle \nabla f(x), \nabla g(u, x)\rangle$$
$$+ \mathbf{1}_{\mathbb{R}^d}(x)(f(x) - f(\infty))g(u, \infty). \quad (22)$$

Using Assumption C.10, we have that for any $u \in [0, t]$

$$\mathbb{E}[\mathbf{1}_{\mathbb{R}^d}(\mathbf{X}_u)\langle \nabla f(\mathbf{X}_u), \nabla g(u, \mathbf{X}_u)\rangle]$$
$$= -\mathbb{E}[\mathbf{1}_{\mathbb{R}^d}(\mathbf{X}_u)g(u, \mathbf{X}_u)(\Delta f(\mathbf{X}_u) + \langle \nabla \log p_u(\mathbf{X}_u), \nabla f(\mathbf{X}_u)\rangle)]. \quad (23)$$

In addition, using Lemma C.9, we have that for any $u \in [0, t]$

$$\mathbb{E}[\mathbf{1}_{\mathbb{R}^d}(\mathbf{X}_u)h(\mathbf{X}_u, \infty)g(u, \infty)] = \mathbb{E}[\mathbf{1}_{\infty}(\mathbf{X}_u) \int_{\mathbb{R}^d} h(x, \mathbf{X}_u)p_u(x)\mathrm{d}x/(1 - S_u)g(u, \mathbf{X}_u)]. \quad (24)$$

Combining (22), (23) and (24), we get that

$$\mathbb{E}[\hat{\mathcal{A}}(fg)(u, \mathbf{X}_u)] = -\mathcal{R}(f)(u, \mathbf{X}_u)g(u, \mathbf{X}_u).$$

In addition, we have

$$\mathbb{E}[(f(\mathbf{X}_t) - f(\mathbf{X}_s))g(\mathbf{X}_t)] = \mathbb{E}[g(t, \mathbf{X}_t)f(\mathbf{X}_t) - f(\mathbf{X}_s)g(s, \mathbf{X}_s)]$$
$$= \mathbb{E}[\int_s^t \mathcal{A}(fg)(u, \mathbf{X}_u)\mathrm{d}u]$$
$$= -\mathbb{E}[\int_s^t \mathcal{R}(f)(u, \mathbf{X}_u)g(u, \mathbf{X}_u)\mathrm{d}u] = -\mathbb{E}[g(\mathbf{X}_t) \int_s^t \mathcal{R}(f)(u, \mathbf{X}_u)\mathrm{d}u],$$

which concludes the proof. $\qquad\square$

Proposition C.11 shows that the time-reversal of a diffusion with killing is a diffusion with the birth. The update can be split into two parts. First, the drift of the diffusion is turned into $(t, x) \mapsto -b(x) + \nabla \log p_{T-t}(x)$. This is in accordance with classical time-reversal results for unconstrained diffusions. The main novelty of Proposition C.11 resides in the change of the killing procedure to a birth procedure. The killing rate $x \mapsto k(x)$ is turned into a birth density $(t, x) \mapsto k(x)p_t(x)/1 - S_t$. This birth density means that, in the time-reversal process, we give birth to a particle near $x$ if $k(x)p_t(x)$ is large. This means that two could conditions must be met for a particle to have a high likelihood to be born (a) the killing rate must be large, i.e. birth can only occur at places where particles died in the original process. (b) the density $p_t$ must be large, i.e. birth can only occur at places which are visited by the original process.

**C.5 Time Reversal of Diffusions with Births.** In the Schrödinger bridge setting, we iterate on the time-reversal procedure. This means that we need to consider the time-reversal of a *birth* process. In this section, we provide the heuristics to derive such a time-reversal. We assume that there exists a birth process $(\mathbf{X}_t)_{t\geq 0}$ taking values in $\hat{\mathbb{R}}^d$ such that $(\mathbf{X}_t)_{t\geq 0}$ is Markov and solution to the martingale problem associated with $\hat{\mathcal{A}}$ where for any $f \in \mathrm{C}^2(\hat{\mathbb{R}}^d)$ and $x \in \hat{\mathbb{R}}^d$ we have

$$\hat{\mathcal{A}}(f)(x) = (\langle b(x), \nabla f(x) \rangle + \tfrac{1}{2}\Delta f(x))\mathbf{1}_{\mathbb{R}^d}(x) + \int_{\mathbb{R}^d}(f(\tilde{x}) - f(\infty))q(\tilde{x})\mathbf{1}_\infty(x).$$

Similarly to the previous section, we consider Assumption C.8 and therefore, we have the existence (and regularity) of a density $p_t$. Under similar assumptions to Proposition C.11, we have that its time-reversal is also the solution to a martingale problem with generator $\hat{\mathcal{R}}$ given for any $f \in \mathrm{C}^2(\hat{\mathbb{R}}^d)$, $t \in [0, T]$ and $x \in \hat{\mathbb{R}}^d$ by

$$\hat{\mathcal{R}}(f)(t, x) = (\langle -b(x) + \nabla \log p_{T-t}(x), \nabla f(x) \rangle + \tfrac{1}{2}\Delta f(x))\mathbf{1}_{\mathbb{R}^d}(x)$$
$$- (1 - S_{T-t})q(x)/p_{T-t}(x)(f(x) - f(\infty))\mathbf{1}_{\mathbb{R}^d}(x). \tag{25}$$

Therefore, the time-reversal of a diffusion process with birth is a diffusion process with death. Similarly to Proposition C.11, the drift is updated as in the unconstrained case, i.e., $x \mapsto b(x)$ is replaced by $(t, x) \mapsto -b(x) + \nabla \log p_{T-t}(x)$. The birth measure $x \mapsto q(x)$ is changed into the killing rate $(t, x) \mapsto (1 - S_{T-t})q(x)/p_{T-t}(x)$.

**Entropic time-reversal with jumps.** In fact (21) and (25) can be inferred from the results of Conforti and Léonard (2022). In this work, the authors consider a pure jump process, i.e., no diffusion, with jumps in $\mathbb{R}^d$. In that case the infinitesimal generator $\mathcal{A}$ is given for any $f \in \mathrm{C}_c^2(\mathbb{R}^d)$ and $x \in \mathbb{R}^d$

$$\mathcal{A}(f)(x) = \langle b(x), \nabla f(x) \rangle + \int_{\mathbb{R}^d}(f(y) - f(x))\mathbb{J}(x, \mathrm{d}y),$$

where $\mathbb{J}$ is a Markov kernel. Under mild entropic assumptions, the authors prove that the time-reversal is also a process with pure jumps associated with the infinitesimal generator $\mathcal{R}$ is given for any $f \in \mathrm{C}_c^2(\mathbb{R}^d)$, $x \in \mathbb{R}^d$ and $t \in [0, T]$ by

$$\mathcal{R}(f)(x) = -\langle b(x), \nabla f(x) \rangle + \int_{\mathbb{R}^d}(f(y) - f(x))\mathbb{R}(x, \mathrm{d}y),$$

where $\mathbb{R}$ is a Markov kernel such that for any $f \in \mathrm{C}_c^2(\mathbb{R}^d \times \mathbb{R}^d)$

$$\int_{\mathbb{R}^d} \int_{\mathbb{R}^d} f(x, y)p_t(x)\mathbb{J}(x, \mathrm{d}y)\mathrm{d}x = \int_{\mathbb{R}^d} \int_{\mathbb{R}^d} f(x, y)p_t(y)\mathbb{R}(y, \mathrm{d}x)\mathrm{d}y.$$

This equation is sometimes called the *flux equation*, see (Conforti and Léonard, 2022, Equation (1.1)). Our setting is different in that we also consider a diffusive part, i.e. there is an additional term $\tfrac{1}{2}\Delta f$ in our generator and our generator is defined on the extended space $\hat{\mathbb{R}}^d$ in order to properly account for the *killing* and *birth* processes.

However, assuming that we can extend the flux equation to our setting, we have in the case of a death process $\mathbb{J}(x, \mathrm{d}y) = k(x)\delta_\infty(\mathrm{d}y)$. Hence, we have that

$$\mathbb{R}(x, \mathrm{d}y)(1 - S_t)\delta_\infty(x) = \mathbb{J}(y, \mathrm{d}x)p_t(y),$$

and therefore $\mathbb{R}(x, \mathrm{d}y) = k(y)p_t(y)/(1 - S_t)\delta_\infty(y)$, which is the update we have in (21). Similarly, in the case of a birth process $\mathbb{J}(x, \mathrm{d}y) = \delta_\infty(x)q(y)\mathrm{d}y$. Hence, we have that

$$\mathbb{R}(x, \mathrm{d}y)p_t(x)\delta_\infty(x) = \mathbb{J}(y, \mathrm{d}x)(1 - S_t),$$

and therefore $\mathbb{R}(x, \mathrm{d}y) = q(x)(1 - S_t)/p_t(x)\delta_\infty(y)$, which is the update we have in (25). Finally, we emphasize that obtaining a ratio of the form $p_t(x)/\mathbb{P}(\mathbf{X}_t = \infty)$ is similar to what has been obtained in the case of *discrete* state-space time-reversal, see (Campbell et al., 2022; Benton et al., 2022) for instance.

**C.6 Duality.** While the formulas in (7) may initially appear intricate and somewhat asymmetric, they are in fact nothing but a pair of *forward* and *backward Kolmogorov* equations on $\hat{\mathbb{R}}^d$: Define $\chi, \hat{\chi} : [0, T] \times \hat{\mathbb{R}}^d \to \mathbb{R}$ as

$$\chi_t(\infty) = \Psi_t, \quad \chi_t(x) = \varphi_t(x) \qquad \text{and} \qquad \hat{\chi}_t(\infty) = \hat{\Psi}_t, \quad \hat{\chi}_t(x) = \hat{\varphi}_t(x).$$

Below, we show that, if $\hat{\mathcal{K}}^{0,\star}$ denotes the *dual* of $\hat{\mathcal{K}}^0$, then (7) is equivalent to

$$\partial_t \chi_t = -\hat{\mathcal{K}}^0 \chi_t, \qquad \partial_t \hat{\chi}_t = \hat{\mathcal{K}}^{0,\star} \hat{\chi}_t.$$

This means that (7) is in fact a system of *dual* equations, even though appearing non-symmetric at first sight. This insight allows us to draw connections to existing research, such as the works by Liu et al. (2022).

Define $\chi, \hat{\chi} : [0, T] \times \hat{\mathbb{R}}^d \to \mathbb{R}$, for any $t \in [0, T]$ and $x \in \hat{\mathbb{R}}^d$, as

$$\chi_t(\infty) = \Psi_t, \quad \chi_t(x) = \varphi_t(x) \qquad \text{and} \qquad \hat{\chi}_t(\infty) = \hat{\Psi}_t, \quad \hat{\chi}_t(x) = \hat{\varphi}_t(x).$$

Recall the formula of a generator with killing:

$$(\hat{\mathcal{K}}^0 f)(x) := \left( \langle b(x), \nabla f(x) \rangle + \frac{1}{2}\Delta f(x) - k(x)\left(f(x) - f(\infty)\right) \right) \mathbf{1}_{\mathbb{R}^d}(x).$$

Then it is evident that the equations governing $\Psi_t$ and $\varphi_t(x)$ can be rewritten as:

$$\partial_t \chi_t = -\hat{\mathcal{K}}^0 \chi_t$$

which is the usual Kolmogorov *backward* equation. On the other hand, we will show below that equations governing $\hat{\Psi}_t$ and $\hat{\varphi}_t(x)$ can be interpreted as the Kolmogorov *forward* equation. To this end, for any two functions $f, g : \hat{\mathbb{R}}^d \to \mathbb{R}$, define the usual $L_2$-inner product as:

$$\langle g, f \rangle := \int_{\mathbb{R}^d} f(x)g(x)\mathrm{d}x + f(\infty) \cdot g(\infty).$$

For any operator $\mathcal{K}$, its dual $\mathcal{K}^\star$ with respect to $\langle \cdot, \cdot \rangle$ is defined as the operator satisfying the relation:

$$\forall f, g, \quad \langle g, \mathcal{K}f \rangle = \langle \mathcal{K}^\star g, f \rangle.$$

Using the standard integration by part formulas, we get

$$\langle g, \hat{\mathcal{K}}^0 f \rangle = \int_{\mathbb{R}^d} g(x)\left( \langle b(x), \nabla f(x) \rangle + \frac{1}{2}\Delta f(x) - k(x)\left(f(x) - f(\infty)\right) \right) \mathrm{d}x$$

$$= \int_{\mathbb{R}^d} \left( \langle g(x)b(x), \nabla f(x) \rangle + \frac{1}{2}g(x)\Delta f(x) - k(x)g(x)f(x) + k(x)g(x)f(\infty) \right) \mathrm{d}x$$

$$= \int_{\mathbb{R}^d} \left( -\mathrm{div}\left(b(x)g(x)\right) + \frac{1}{2}\Delta g(x) - k(x)g(x) \right) f(x)\mathrm{d}x + \left( \int_{\mathbb{R}^d} k(x)g(x)\mathrm{d}x \right) \cdot f(\infty)$$

$$=: \langle \hat{\mathcal{K}}^{0,\star} g, f \rangle$$

where the dual of $\hat{\mathcal{K}}^0$ is given by

$$(\hat{\mathcal{K}}^{0,\star} g)(x) := \left( -\mathrm{div}\left(b(x)g(x)\right) + \frac{1}{2}\Delta g(x) - k(x)g(x) \right) \mathbf{1}_{\mathbb{R}^d} + \left( \int_{\mathbb{R}^d} k(x)g(x)\mathrm{d}x \right) \mathbf{1}_\infty.$$

With these definitions, it holds that $\hat{\chi}$ satisfies the *Forward Kolmogorov* equation: (7) is equivalent to

$$\partial_t \hat{\chi}_t = \hat{\mathcal{K}}^{0,\star} \hat{\chi}_t.$$

## D   UNBALANCED IPF

In this section, we state the SDEs describing the evolution of log-potentials $\log \varphi$ and $\log \hat{\varphi}$ along backward and forward trajectories of the optimal solution to the unbalanced SB problem. We then further comment on the sampling strategies used for birth processes. We provide formulas which generalize the prior process (3) to diffusions with arbitrary (scalar) diffusivity $\sigma : [0, 1] \to \mathbb{R}_+$.

**D.1  SDEs for Log-Potentials.**   To simplify the notation, in this and subsequent sections, we denote the Neural Networks parametrizing $\log \varphi$ and $\log \hat{\varphi}$ as $Y^\theta$ and $\hat{Y}^{\hat{\theta}}$, and their score as:

$$Z^\theta(t, x_t) = \sigma \nabla_{\mathbf{x}} Y^\theta(i, x_t)$$
$$\hat{Z}^{\hat{\theta}}(t, x_t) = \sigma \nabla_{\mathbf{x}} \hat{Y}^{\hat{\theta}}(i, x_t)$$

**Forward Dynamics**   When accounting for time-varying diffusivity, the SDE part of the forward diffusion process (3) reads:

$$\mathrm{d}\overrightarrow{X}_t = (b + \sigma Z^\theta)\,\mathrm{d}t + \sigma\,\mathrm{d}W_t \tag{26}$$

Along paths $(\overrightarrow{X}_t)_t$, the log-potentials evolve as:

$$\begin{cases} \mathrm{d}Y_t = \left( \frac{1}{2}\|Z_t\|^2 + k_t\left(1 - \frac{\Psi}{\exp Y_t}\right) \right)\mathrm{d}t + \langle Z_t,\, \mathrm{d}W_t \rangle \\ \mathrm{d}\hat{Y}_t = \left( \frac{1}{2}\left\|\hat{Z}_t\right\|^2 + \nabla_{\mathbf{x}} \cdot (-b + \sigma \hat{Z}_t) + \left\langle \hat{Z}_t,\, Z_t \right\rangle - k_t \right)\mathrm{d}t + \left\langle \hat{Z}_t,\, \mathrm{d}W_t \right\rangle \end{cases} \tag{27}$$

**Backward Dynamics**   The time-reversal of (26) is:

$$\mathrm{d}\overleftarrow{X}_t = (b - \sigma \hat{Z}^{\hat{\theta}})\,\mathrm{d}t + \sigma\,\mathrm{d}W_t \tag{28}$$

The log-potentials along $(\overleftarrow{X}_t)_t$ follow, instead, the SDEs:

$$\begin{cases} \mathrm{d}Y_s = \left( \frac{1}{2}\|Z_s\|^2 + \sigma \nabla_{\mathbf{x}} \cdot Z_s + \left\langle Z_s,\, \hat{Z}_s \right\rangle + k_s\left(\frac{\Psi}{\exp Y_s} - 1\right) \right)\mathrm{d}s + \langle Z_s,\, \mathrm{d}W_s \rangle \\ \mathrm{d}\hat{Y}_s = \left( \nabla_{\mathbf{x}} \cdot b + \frac{1}{2}\left\|\hat{Z}_s\right\|^2 + k_s \right)\mathrm{d}s + \left\langle \hat{Z}_s,\, \mathrm{d}W_s \right\rangle \end{cases} \tag{29}$$

**D.2  Discretization of Diffusions with Killing and Birth.**   In Algorithm 2, we need to sample both from the forward (function SAMPLE-FORWARD) and backward dynamics (function SAMPLE-BACKWARD). We discretize Eqs. (26) and (28) using the standard Euler-Maruyama discretization ($i \in \{0, I\}$ is the discrete time index):

$$\Delta \overrightarrow{X}_i = (b + \sigma Z^\theta)\Delta t + \sigma \Delta W_i$$

and

$$\Delta \overleftarrow{X}_i = (b - \sigma \hat{Z}^{\hat{\theta}})\Delta t + \sigma \Delta W_i$$

The quantities $\Delta \hat{Y}$ and $\Delta Y$ are obtained analogously by discretizing Eqs. (27.2) and (29.1).

The exact mechanism simulating transitions to and from the coffin state is provided in Algorithm 2. The function Sample-Backward uses shadow paths (see §4) and a rate equal to $\Psi \frac{k(i, X_i)}{R \exp Y^\theta(i, X_i)}\Delta t$.

This can be obtained by setting $\gamma = \Delta t$ and recalling the identity $\Psi_t \hat{\Psi}_t = 1 - \int p_t(x)\mathrm{d}x$. The only discrepancy between our implementation and birth sampling in Section 4 is that we do not sample reverse-time trajectories, to define the birth locations of shadow particles. This simplifies training, during which we need to sample trajectories in one direction in order to update the drift pertaining to the other direction: The advantage consists in not using the networks we are learning to sample paths.

## E    RELATED WORK

**Unbalanced optimal transport.**    We start by briefly recalling two different approaches to unbalanced OT. First, the *hard* marginal constraints can be relaxed and replaced by *soft* ones. We refer to Chizat et al. (2018); Liero et al. (2018); Yang and Uhler (2019); Kondratyev et al. (2016) and the references therein. Second, the measures of interest can be extended to satisfy the *conservation of mass* constraint. This is usually done by adding a *coffin* state $\{\infty\}$. After this operation, it is possible to apply standard (although defined on this extended space) tools from OT (Pele and Werman, 2009; Caffarelli and McCann, 2010; Gramfort et al., 2015; Ekeland, 2010). These works also include entropic relaxations of the OT objective. Moving to the dynamic and static formulations of unbalanced Schrödinger bridges, Chen et al. (2022) have obtained the main properties of the Schrödinger Bridge in the unbalanced setting. To the best of our knowledge, our study of the iterates of the Iterate Proportional Fitting procedure is new.

**Diffusion models.**    The time reversal of the continuous part of the dynamics has been investigated in Anderson (1982); Haussmann and Pardoux (1986); Cattiaux et al. (2021). It is at the basis of diffusion models (Ho et al., 2020; Song et al., 2021a). Such approaches represent the state-of-the-art generative models in many domains, from text-to-image generation (Saharia et al., 2022) to protein modeling (Watson et al., 2022). To the best of our knowledge, no existing diffusion model includes birth and death mechanisms in the forward and backward processes. Hence, the results we obtain are also new from a diffusion model perspective. We highlight that the updates we obtain on the birth and death rates resemble closely the update obtained for the time-reversal of discrete state-space diffusion models such as Shi et al. (2023). We refer to Benton et al. (2022) for a treatment of general diffusion models through the lens of the infinitesimal generator. An alternative approach to our derivations would have been to use these results to establish (i) a variational lower-bound defining a loss function, (ii) a time-reversal formula. We leave such a study for future work.

**Diffusion Schrödinger bridges and extensions.**    The Diffusion Schrödinger Bridge (DSB) (De Bortoli et al., 2021; Vargas et al., 2021b; Chen et al., 2021) is a new paradigm to solve transport problems efficiently in high dimensions. It relies on the advent of diffusion models. Several extensions of DSBs have been proposed. Closest to our work is Liu et al. (2022), which studies a *generalized* version of the SB problem. In that case, an extra functional is added to the quadratic control corresponding to the Schrödinger bridge. Even though the updated equations share some similarities with ours, we highlight some key differences: (i) there is an extra term in the Partial Differential Equation system we consider, which can be identified with the fact that we consider *death*, (ii) the dynamics in Liu et al. (2022) do not include birth and death mechanisms. As a result, the obtained procedure corresponds to a numerical scheme for Chen et al. (2022, Section 5), which describes a *reweighted* SB. While the reweighted approach can also account for the loss of mass, it does not update the particles in a meaningful way from a transport point of view, since the death and birth rate are never updated. Theoretical properties of the solutions to Schrödinger bridge problems are investigated in Chen et al. (2022). Finally, we highlight that recent approaches (Somnath et al., 2023; Liu et al., 2023; Shi et al., 2023) introduce new schemes to solve SBs based on the ideas developed in Lipman et al. (2022).

## F    ALGORITHMIC DETAILS

In this section, we clarify the design of our algorithm UDSB-F, which we introduce to mitigate the issues of UDSB-TD. We present an approximation to the computation of the posterior killing rate which is conducive to stability and scalability. We highlight the changes to the training and sampling procedures that this approximation entails and how it allows us to tackle forward dynamics with both killing and births.

**F.1    Estimating Extremal values of Log-Potentials.**    Sampling procedures in Algorithm 2 iteratively compute the values of $Y$ and $\hat{Y}$ by starting from one extreme (either $\hat{Y}_0$ or $Y_I$) and then adding (or subtracting) the increments $\Delta Y$ and $\Delta \hat{Y}$ at each time-step. The computation of $\hat{Y}_0$ or $Y_I$, however,

requires some care. These extreme values can be approximated by:

$$\hat{Y}_0 = \log \frac{\mu_0}{\varphi_0} \approx \log \mu_0 - Y_0^\theta$$

$$Y_I = \log \frac{\mu_1}{\hat{\varphi}_I} \approx \log \mu_1 - \hat{Y}_I^{\hat{\theta}},$$

but to evaluate these expressions we need access to the log-densities $\log \mu_0$ and $\log \mu_1$. Obviously, we should not rely on the availability of exact, closed-form expressions, since that would make our algorithm not applicable to empirical data. We, therefore, resort to coarse approximations computed once before starting the training. We can, for instance, fit Bayesian Gaussian Mixture models but different solutions may be preferable when working with specific types of data.

We note that the need for estimates of marginal densities is a burdensome requirement, which is not placed by solvers of balanced SBs and its removal motivates our algorithm UDSB-F.

Lastly, we remark that when sampling from the backward time direction (as in `Sample-Backward`), we should account for births (rather than deaths) since, as described in Proposition 3.1, the time-reversal of a death process is a birth process.

---

**Algorithm 2** UDSB-TD sampling (death forward process)

---

**Input:** drift $b$, diffusivity $\sigma$, initial mass $M_0$, final mass $M_1$, killing function $k$

1: **function** SAMPLE-FORWARD($Y^\theta, \hat{Y}^{\hat{\theta}}, \Psi$)
2:     Sample position $X_0 \sim \mu_0$
3:     $A_0 \leftarrow 1$                                                                  ▷ *The particle is initially alive*
4:     $\hat{Y}_0 \leftarrow$ COMPUTE-INITIAL-Y-HAT($Y^\theta, \hat{Y}^{\hat{\theta}}$)
5:     **for** step $i$ in $\{1, ..., I\}$ **do**
6:         $X_i \leftarrow X_{i-1} + \Delta\vec{X}_i$
7:         $\hat{Y}_i \leftarrow \hat{Y}_{i-1} + \Delta\hat{Y}_i$
8:         **if** $A_{i-1} = 1$ **then**                                       ▷ *The particle is alive*
9:             Flip coin $D \sim$ Bernoulli $\left(1 - \Psi\frac{k(i,X_i)}{\exp Y^\theta(i,X_i)}\Delta t\right)$
10:             $A_{i+1} \leftarrow D$                                      ▷ *Store particle's fate*
11:     **return** $(X_i, \hat{Y}_i, A_i)_i$
12:
13: **function** SAMPLE-BACKWARD($Y^\theta, \hat{Y}^{\hat{\theta}}, \Psi$)
14:     Sample position $X_I \sim \mu_1$
15:     Sample status $A_I \sim$ Bernoulli $\left(\min\left(1, \frac{M_1}{M_0}\right)\right)$        ▷ *Is the particle alive at $k = K$?*
16:     $Y_I \leftarrow$ COMPUTE-INITIAL-Y($Y^\theta, \hat{Y}^{\hat{\theta}}$)
17:     **for** step $i$ in $\{I-1, ..., 0\}$ **do**
18:         $X_i \leftarrow X_{i+1} - \Delta\overleftarrow{X}_i$
19:         $Y_i \leftarrow Y_{i+1} - \Delta Y_i$
20:         **if** $A_{i+1} = 0$ **then**                                       ▷ *The particle is dead*
21:             $R \leftarrow$ Fraction of dead particles at time $i + 1$:
22:             Flip coin $D \sim$ Bernoulli $\left(\Psi\frac{k(i,X_i)}{R \exp Y^\theta(i,X_i)}\Delta t\right)$
23:             $A_i \leftarrow D$                                       ▷ *Store particle's fate*
24:     **return** $(X_i, Y_i, A_i)_i$

---

**F.2 Training Objectives.** In this section, we explicitly state the training objectives used by UDSB-TD.

### F.2.1 MM LOSSES

**Mean-Matching Objective.** It is easy to adapt the *mean-matching* objective (De Bortoli et al., 2021) to UDSBs since it is computed from paths of live particles –which follow a standard diffusion process–

and is therefore clearly independent from deaths. We use the following first-order approximation (in $\Delta t$) of its original expression ($\tilde{b}$ represents indifferently any forward or backward drift):

$$\mathcal{L}_{\text{mm}} \approx \Delta t \sum_i \mathbb{E}_{X_i} \left[ \left\| (X_{i+1} - X_i) - \tilde{b}\Delta t \right\|^2 \right].$$

When evaluated on forward $(\overrightarrow{X}_i)_i$ and backward $(\overleftarrow{X}_i)_i$ paths, this loss becomes:

$$\mathcal{L}_{\text{mm}}((\overleftarrow{X}_i)_i; \theta) = \Delta t \sum_i \left\| \left( \overleftarrow{X}_{i+1} - \overleftarrow{X}_i \right) - (b + \sigma^2 \nabla_{\mathbf{x}} Y^\theta)\Delta t \right\|^2$$

$$\mathcal{L}_{\text{mm}}((\overrightarrow{X}_i)_i; \hat{\theta}) = \Delta t \sum_i \left\| \left( \overrightarrow{X}_{i+1} - \overrightarrow{X}_i \right) - (b - \sigma^2 \nabla_{\mathbf{x}} \hat{Y}^{\hat{\theta}})\Delta t \right\|^2.$$

Minimizing these quantities allows, in our case, to learn the forward/backward trajectories of particles that do not jump to the coffin state. However, it is not sufficient to fully characterize the magnitude of $Y^\theta$ and $\hat{Y}^{\hat{\theta}}$, since only their gradients appear in it. $\mathcal{L}_{\text{mm}}$ is in fact invariant to constant shifts: i.e., there is no difference between $Y^\theta$ and $(Y^\theta - c)$ during training.

**Divergence Objective.** An alternative $\mathcal{L}_{\text{MM}}$ candidate is given by the *divergence-based objective* $\mathcal{L}_{\text{div}}$ by Vargas et al. (2021a), which also appear in Chen et al. (2021). Its original formulation in the context of balanced SBs depends only on the scores $Z$ and $\hat{Z}$. Surprisingly, the same is not true for USBs since this objective is affected by the presence of deaths. We extend it to the unbalanced case by recalling (Liu et al., 2022) that $\mathcal{L}_{\text{div}}$ can be also expressed as:

$$\mathcal{L}_{\text{div}}(\theta) = \mathbb{E}_{X_s \sim \overleftarrow{X}} \left[ \int_0^1 \left( \mathrm{d}Y_s^\theta(X_s) + \mathrm{d}\hat{Y}_s^{\hat{\theta}}(X_s) \right) \right]$$

$$\mathcal{L}_{\text{div}}(\hat{\theta}) = \mathbb{E}_{X_t \sim \overrightarrow{X}} \left[ \int_0^1 \left( \mathrm{d}Y_t^\theta(X_t) + \mathrm{d}\hat{Y}_t^{\hat{\theta}}(X_t) \right) \right],$$

where the first expectation runs over the backward dynamics (from $N$) while the second uses the forward dynamics (from $Q$). The terms related to jumps that appear in the differentials of $\hat{Y}$ and $Y$ (Eqs. 27 and 29) do not completely cancel out when summed (i.e., the term $k\Psi$ from $\mathrm{d}Y$ remains). The objective $\mathcal{L}_{\text{div}}(\theta)$, computed as the sum of the equations in (29), is therefore:

$$\mathcal{L}_{\text{div}}(\theta) = \int_0^1 \mathbb{E}_{X_s \sim \overleftarrow{X}} \left[ \frac{1}{2} \left\| Z_s^\theta(X_s) \right\|^2 + \sigma \nabla_{\mathbf{x}} \cdot Z_s^\theta(X_s) + \left\langle Z_s^\theta(X_s), \hat{Z}_s^{\hat{\theta}}(X_s) \right\rangle - \frac{k\Psi}{e^{Y_s^\theta(X_s)}} \right] \mathrm{d}s,$$

in which the term in blue is unique to USBs. Note that this new death-related term depends on the parameter that is being optimized ($\theta$) and cannot, therefore, be ignored when training the network $Y^\theta$. The same term also appears in the objective $\mathcal{L}_{\text{div}}(\hat{\theta})$, which is obtained by summing the differentials in (27):

$$\mathcal{L}_{\text{div}}(\hat{\theta}) = \int_0^1 \mathbb{E}_{X_t \sim \overrightarrow{X}} \left[ \frac{1}{2} \left\| \hat{Z}_t^{\hat{\theta}}(X_t) \right\|^2 + \sigma \nabla_{\mathbf{x}} \cdot \hat{Z}_t^{\hat{\theta}}(X_t) + \left\langle \hat{Z}_t^{\hat{\theta}}(X_t), Z_t^\theta(X_t) \right\rangle - \frac{k\Psi}{e^{Y_{1-t}^\theta(X_t)}} \right] \mathrm{d}t.$$

In this case, however, the blue quantity is irrelevant since it does not depend on $\hat{\theta}$. By discretizing the two expressions for $\mathcal{L}_{\text{div}}$ and removing unnecessary terms we obtain the losses:

$$\mathcal{L}_{\text{div}}\left( (\overleftarrow{X}_i)_i; \theta \right) = \Delta t \sum_i \left( \frac{1}{2} \left\| Z_i^\theta(\overleftarrow{X}_i) \right\|^2 + \sigma \nabla_{\mathbf{x}} \cdot Z_i^\theta(\overleftarrow{X}_i) \right.$$

$$\left. + \left\langle Z_i^\theta(\overleftarrow{X}_i), \hat{Z}_i^{\hat{\theta}}(\overleftarrow{X}_i) \right\rangle - \frac{k\Psi}{e^{Y_i^\theta(\overleftarrow{X}_i)}} \right) \tag{30}$$

$$\mathcal{L}_{\text{div}}\left( (\overrightarrow{X}_i)_i; \hat{\theta} \right) = \Delta t \sum_i \left( \frac{1}{2} \left\| \hat{Z}_i^{\hat{\theta}}(\overrightarrow{X}_i) \right\|^2 + \sigma \nabla_{\mathbf{x}} \cdot \hat{Z}_i^{\hat{\theta}}(\overrightarrow{X}_i) + \left\langle \hat{Z}_i^{\hat{\theta}}(\overrightarrow{X}_i), Z_i^\theta(\overrightarrow{X}_i) \right\rangle \right).$$

**Choosing the MM loss.**   Either one of the losses $\mathcal{L}_{\mathrm{mm}}$ and $\mathcal{L}_{\mathrm{div}}$ presented above can be used as $\mathcal{L}_{\mathrm{MM}}$ in Algorithm 1. However, the choice determines the type of admissible reference drifts –which must be linear if the mean-matching loss $\mathcal{L}_{\mathrm{mm}}$ is selected– and the computational cost of training –which is higher for $\mathcal{L}_{\mathrm{div}}$ due to the computation of the divergence.

### F.2.2   TD Loss

While sufficient to ensure that $\nabla_{\mathbf{x}} Y^{\theta}$ and $\nabla_{\mathbf{x}} \hat{Y}^{\hat{\theta}}$ point "in the right direction", $\mathcal{L}_{\mathrm{MM}}$ objectives described in the previous section do not constrain the magnitude of $\hat{Y}^{\hat{\theta}}$ (which always appears as $\hat{Z}^{\hat{\theta}} = \sigma \nabla \hat{Y}^{\hat{\theta}}$). Furthermore, the only objective ($\mathcal{L}_{\mathrm{div}}(\theta)$) which depends on the magnitude of $Y^{\theta}$ is likely insufficient to learn it in practice: the blue term in (30) disappears when the prior killing rate is 0 or when $\Psi$ becomes exceedingly small, leading to a very weak signal on $Y^{\theta}$. This motivates our quest for a second type of loss ($\mathcal{L}_{\mathrm{TD}}$) which allows learning the magnitudes of $Y^{\theta}$ and $\hat{Y}^{\hat{\theta}}$.

We adopt the TD loss introduced by Liu et al. (2022). Its expressions for both networks are given by:

$$\mathcal{L}_{\mathrm{TD}}\left(\theta\right) = \int_0^1 \mathbb{E}_{X_s \sim \overleftarrow{X}} \left[ |Y_s^{\theta}(X_s) - Y_s| \right] \mathrm{d}s$$

$$\mathcal{L}_{\mathrm{TD}}\left(\hat{\theta}\right) = \int_0^1 \mathbb{E}_{X_t \sim \overrightarrow{X}} \left[ |\hat{Y}_t^{\hat{\theta}}(X_t) - \hat{Y}_t| \right] \mathrm{d}t,$$

where $(Y_s)_s$ is computed using (29) and $(\hat{Y}_t)_t$ using (27). In words, the TD loss measures the L1 distance between the values of $Y$ and $\hat{Y}$ –which we compute on paths from $N$ and $Q$ respectively– and the output of neural networks $Y^{\theta}$ and $\hat{Y}^{\hat{\theta}}$.

**F.3   Computing $\Psi$.**   The previous section describes the losses designed to optimize $Y^{\theta}$ and $\hat{Y}^{\hat{\theta}}$ and we now describe how to update $\Psi$, which is the third quantity estimated by UDSB-TD (being required to compute the posterior killing rate $\Psi k / \varphi$). Its value is refreshed at the beginning of every training epoch (line 2 of Algorithm 1) by UPDATE-PSI(). $\Psi$ can be computed from the closed-form expression:

$$\Psi = \left( \int_{\mathcal{X}} \mu_0 \, \mathrm{d}x - \int_{\mathcal{X}} \mu_1 \, \mathrm{d}x \right) \left( \int_0^1 \mathrm{d}t \int_d \mathrm{d}x \, k \hat{\varphi}(t, x) \right)^{-1} \tag{31}$$

which directly follows from the Schrödinger Problem [Chen Unbalanced]. We use the following two approximations.

**Approximation 1**   From (31), by multiplying and dividing by $\varphi$ inside the integral and then applying the decomposition $\varphi_t \hat{\varphi}_t = p_t$, we obtain:

$$\Psi = (m_0 - m_1) \left( \int_0^1 \mathrm{d}t \int_d \mathrm{d}x \, \frac{k}{\varphi} \hat{\varphi} \varphi(t, x) \right)^{-1}$$

$$= (m_0 - m_1) \left( \int_0^1 \mathrm{d}t \, \mathbb{E}_{X_t} \left[ \frac{k}{\varphi}(t, X_t) \right] \right)^{-1}$$

$$\approx (m_0 - m_1) \left( \sum_i \frac{\Delta t}{N} \sum_n \frac{k(i\Delta t, X_i^n)}{\exp Y^{\theta}(i\Delta t, X_i^n)} \right)^{-1},$$

In theory, this estimator of $\Psi$ is non-biased only when the HJB equation is satisfied, i.e., when the TD loss is minimized, since only then $\varphi_t \hat{\varphi}_t = p_t$ holds. We however argue that this does not disrupt training. In particular, we empirically found that it is helpful to introduce a warm-up period in training, during which only the MM loss is optimized and $\Psi$ is set to 0. This can be intuitively explained by noting that, at the beginning, trajectories will likely visit the wrong parts of the space, i.e., regions that in reality have almost zero density (i.e., $\varphi \hat{\varphi} \approx 0$). Minimizing the TD loss over such paths would therefore yield very bad results. At a later stage, instead, trajectories are supposed to be approximately correct. We are therefore gradually able to reintroduce the TD loss, which (as seen in the proof of Liu et al. (2022, Prop. 4)) ensures the respect of HJB, and also update $\Psi$.

**Approximation 2** To derive a second heuristic which uses the network $\hat{Y}^{\hat{\theta}}$ rather than $Y^\theta$, we replace integrals over $\mathbb{R}^d$ in (31) with sums along paths $(X_i)_i$:

$$\Psi \approx (m_0 - m_1) \left( \sum_i \Delta t \sum_n k(i\Delta t, X_i^n) \exp \hat{Y}^{\hat{\theta}}(i\Delta t, X_i^n) \right)^{-1}.$$

In the above formula, the superscript $n$ indexes members of the batch. We stress that this approximation is biased and may suffer from high variance, but we include it here only because of its practical interest in some situations.

**F.4 Revising the Update of $\Psi$.** The above formulas to update $\Psi$ are heavily dependent on good guesses of $Y^\theta$ and $\hat{Y}^{\hat{\theta}}$, since they both appear inside exponential terms. In practice, however, reliable estimates may not always be available, for several reasons. First, the user-defined approximations of $Y$ and $\hat{Y}$ at the extremes may not be good, especially with high dimensional state spaces and complex data distributions. Second, even if learned approximations of log-potentials $\varphi$ and $\hat{\varphi}$ improve by the end of training, they can still be inadequate at the beginning.

Unreliable values of $Y^\theta$ and $\hat{Y}^{\hat{\theta}}$ hamper convergence, making it slower or, in the worst case, precluding it. We, therefore, re-parametrize the posterior killing rate $k^\star = k\Psi/\varphi$ with the help of an additional neural network $g^\zeta$ as:

$$k^\star \approx k(t, x) g_t^\zeta(x).$$

We learn $g^\zeta$ using a novel loss, called *Ferryman* loss, which we present next.

**The Ferryman loss.** At equilibrium, the amount of mass that reaches the coffin state $\infty$ is given by:

$$\int_0^1 \int_{\mathbb{R}^d} k^\star(t, x) p_t(x) \, \mathrm{d}x \, \mathrm{d}t$$

and should be equal to the difference of mass observed at the marginals $\mu_0(\mathbb{R}^d) - \mu_1(\mathbb{R}^d)$. By matching these two expressions, we get the loss:

$$\mathcal{L}(\zeta) = \int_0^1 \mathbb{E}\left[ k(\mathbf{X}_t) g_{\zeta,t}(\mathbf{X}_t) \right] \mathrm{d}t - \mu_1(\mathbb{R}^d) + \mu_0(\mathbb{R}^d), \tag{32}$$

where the expectation runs over paths. We stress that this loss does not guarantee the convergence of $kg^\zeta$ to the optimal form of posterior killing rate $k\Psi/\varphi$ but achieves good-enough results in practice.

The loss function (32) tries to capture a mass constraint at the extremes, i.e., for $t = 0$ and $t = 1$. However, we can extend it to the more general case in which the size of the population $m_t$ is also known at several intermediate times $t_i, i \in I$. This increases the amount of information that USBs are able to incorporate and allows learning better dynamics in many real-world applications.

We call $(X_t)_t$ a trajectory and define binary random variables $A_t$ which specify the status of the particle, by taking the value 1 when $X_t \neq \infty$. For the set of mass measurements $m_t$, it should hold that $\mathbb{P}(A_t = 1) = \mathbb{E}_{A_t}[A_t] = m_t$. This constraint is, however, ill-suited to be directly optimized, since $A_t$ is a discrete value and we should back-propagate through the expected value. We, therefore, replace it with the quantity $\mathbb{E}_{(X_t, A_t)}\left[ k^\zeta(t, X_t) \mathbf{1}\{A_t = 1\} \right]$. Furthermore, we relax the requirement that the prior process is a death-only diffusion and consider a prior birth function $q$, together with the usual killing function $k$.

The loss function used to train $g^\zeta$ then becomes:

$$\mathcal{L}_F(\zeta) = \mathbb{E}_{(X_t, A_t)} \left[ \sum_{i \in I} \left| \int_0^{t_i} \left( (1 - A_t) q g^\zeta(t) - A_t k g^\zeta(t) \right) \mathrm{d}t - \frac{m_{t_i} - m_0}{M} \right| \right], \tag{33}$$

where the dependency of $k$, $q$ and $g^\zeta$ on $X_t$ is hidden and $M = \max_i m_{t_i}$ is a normalization constant. In words, $\mathcal{L}_F$ computes a soft count of deaths and births at each timestep $t$, by summing the probabilities of sampled particles to transition from/to the coffin state. Clearly, these transition probabilities depend on the status $A_t$ of each particle since, e.g., only alive particles can die. For each time interval $[0, t_i]$, $\mathcal{L}_F$ ensures that the approximated change of mass stays close to the observed variation $(m_{t_i} - m_0)/M$.

To use (33) during training, we should also prevent the value of $g^\zeta$ from exploding. In our experiments, we, therefore, use a regularized version of this loss, which we illustrate next.

**Discretizing the Ferryman loss.** The loss (33) may be unstable in practice because $g^\zeta$ can grow without bounds. This would translate, upon discretization, into invalid transition probabilities $k\Delta t$ to the coffin state, i.e., probabilities bigger than 1. We, therefore, modify the loss to prevent this. We define the following quantities:

$$k_{\to\infty}(x_n) = \mathbb{P}\left(X_{n+1} = \infty | X_n = x_n\right) = kg^\zeta(n, x_n)\Delta t$$

$$q_{\leftarrow\infty}(x_n) = \mathbb{P}\left(X_{n+1} = x_n | X_n = \infty\right) = qg^\zeta(n, x_{n+1})\Delta t,$$

where $(x_n)_n$ is a sampled trajectory and which are, respectively, the conditional probabilities of death ($k_{\to\infty}$) and birth ($q_{\leftarrow\infty}$) given the position at step $n$. The revised Ferryman loss can then be written as:

$$\mathcal{L}_F(\zeta) = \mathbb{E}_{(X_t, A_t)}\left[\sum_{i\in I}\left|\sum_{n=0}^{n_i}\left((1 - A_n)\lceil q_{\leftarrow\infty}(X_n)\rfloor - A_n\lceil k_{\to\infty}(X_n)\rfloor\right) - \frac{m_{n_i} - m_0}{M}\right| + \right.$$
$$\left. + \sum_{n=0}^{n_i}\left|q_{\leftarrow\infty}(X_n) - \lceil q_{\leftarrow\infty}(X_n)\rfloor\right| + \sum_{n=0}^{n_i}\left|k_{\to\infty}(X_n) - \lceil k_{\to\infty}(X_n)\rfloor\right|\right], \tag{34}$$

where $\lceil x\rfloor$ denotes clipping of $x$ to the unit interval. (34) differs in two ways from (33): (i) it uses clipped transition probabilities in the first sum and (ii) contains two regularization terms, which penalize values of $q_{\leftarrow\infty}$ and $k_{\to\infty}$ bigger than 1.

**F.5 A Revised Algorithm.** We detail the revised training procedure of UDSB-F in Algorithm 3. The associated sampling procedures are given in Algorithm 4 in the specific case of a death-only forward process, to allow a direct comparison with the original sampling in Algorithm 2.

---

**Algorithm 3** UDSB-F training

---

**Input:** $Z^\theta, \hat{Z}^{\hat\theta}, g^\zeta$
**Output:** $\theta, \hat\theta, \zeta$

1: **for** epoch $e \in \{0, ..., E\}$ **do**
2:     $(\overleftarrow{X}_i, A_i)_i \leftarrow$ SAMPLE-BACKWARD-F$(Z^\theta, \hat{Z}^{\hat\theta}, g^\zeta)$
3:     **while** reuse paths **do**
4:         $L_{\text{MM}}(\theta) \leftarrow \mathcal{L}_{\text{MM}}\left((\overleftarrow{X}_i); \theta\right)$
5:         Update $\theta$ using $\nabla_\theta L_{\text{MM}}$                     ▷ *Train forward score*
6:     $(\overrightarrow{X}_i, A_i)_i \leftarrow$ SAMPLE-FORWARD-F$(Z^\theta, \hat{Z}^{\hat\theta}, g^\zeta)$
7:     **while** reuse paths **do**
8:         $L_{\text{MM}}(\hat\theta) \leftarrow \mathcal{L}_{\text{MM}}\left((\overrightarrow{X}_i); \hat\theta\right)$
9:         Update $\hat\theta$ using $\nabla_{\hat\theta} L_{\text{MM}}$                     ▷ *Train backward score*
10:         $L_F(\zeta) \leftarrow \mathcal{L}_F\left((\overrightarrow{X}_i, A_i); \zeta\right)$
11:         Update $\zeta$ using $\nabla_\zeta L_F$                     ▷ *Train transition function*

---

---

**Algorithm 4** UDSB-F sampling (death-only prior)

---

**Input:** drift $b$, diffusivity $\sigma$, initial mass $M_0$, final mass $M_1$, killing function $k$

1: **function** SAMPLE-FORWARD-F$(Z^\theta, \hat{Z}^{\hat{\theta}}, g^\zeta)$
2:      Sample position $X_0 \sim \mu_0$
3:      $A_0 \leftarrow 1$                                 ▷ *The particle is initially alive*
4:      **for** step $i$ in $\{1, ..., I\}$ **do**
5:          $X_i \leftarrow X_{i-1} + \overrightarrow{\Delta X_i}$
6:          **if** $A_{i-1} = 1$ **then**                       ▷ *The particle is alive*
7:              Flip coin $D \sim$ Bernoulli $\left(1 - k(i, X_i)g^\zeta(i)\Delta t\right)$
8:              $A_{i+1} \leftarrow D$                         ▷ *Store particle's fate*
9:      **return** $(X_i, A_i)_i$
10:
11: **function** SAMPLE-BACKWARD-F$(Z^\theta, \hat{Z}^{\hat{\theta}}, g^\zeta)$
12:      Sample position $X_I \sim \mu_1$
13:      Sample status $A_I \sim$ Bernoulli $\left(\min\left(1, \frac{M_1}{M_0}\right)\right)$       ▷ *Is the particle alive at $k = K$?*
14:      **for** step $i$ in $\{I-1, ..., 0\}$ **do**
15:          $X_i \leftarrow X_{i+1} - \overleftarrow{\Delta X_i}$
16:          **if** $A_{i+1} = 0$ **then**                       ▷ *The particle is dead*
17:              Flip coin $D \sim$ Bernoulli $\left(k(i, X_i)g^\zeta(i)\Delta t\right)$
18:              $A_i \leftarrow D$                           ▷ *Store particle's fate*
19:      **return** $(X_i, A_i)_i$

---

## G    EXPERIMENTS

In this section, we provide further details on the model architecture, training parameters, and metrics used. We also discuss two sets of additional experiments: The first (§G.1) compares the performances of the two unbalanced Schrödinger bridge (USB) solvers (both on synthetic and real data). The second (§G.4) involves modeling the emergence of the Delta variant during the COVID pandemic.

**G.1 Comparison between UDSB-TD and UDSB-T.** We aim to test the ability of our two solvers, UDSB-TD and UDSB-F, to compute valid SBs while respecting arbitrary mass constraints. To this end, we run both algorithms on the 2-dimensional toy data displayed in Fig. 5a and then on a second dataset involving cellular dynamics.

**Toy dataset.** Points are initially drawn from a mixture of (i) a uniform distribution (left, *blue* segment) and (ii) an isotropic Gaussian (center). Their final distribution is instead uniform and supported on two segments (right, in *red*). Given the constraints on their endpoints, particles –which follow a Brownian prior– should travel from left to right. In the process, however, they cross the area marked with a rectangle in the picture. It denotes a region of the state space in which deaths can occur, i.e., a region in which $k$ is non-zero.

We test our SB solvers under multiple mass scenarios, by specifying four different amounts of live particles at the end (Fig. 5b). We picture the trajectories found by UDSB-TD in Fig. 5c, and those computed by UDSB-F in Fig. 5d. We observe that, in both cases, the paths constitute valid SBs, i.e., correctly match the marginals. Furthermore, the predicted end distributions are similar in quality in all scenarios (Table 2), and no algorithm consistently achieves better scores.

The trajectories computed by UDSB-TD and UDSB-F are nevertheless not identical. In particular, differences emerge when comparing the positions of deaths (*black* dots), which are concentrated around the top and bottom ends of the killing region in Fig. 5c, while appearing more uniform in Fig. 5d. This discrepancy is a direct consequence of different ways of computing $\Psi$.

UDSB-TD updates $\Psi$ using formulas that depend on log-potentials ($\log\varphi$ and $\log\hat{\varphi}$) which, as discussed in §F.4, are usually not well-approximated at the beginning. Inaccurate values of potentials lead, in the initial phase of training, to excessive deaths and therefore push the trajectories away from the death region, i.e., drive particles above or below the rectangle. When the estimates of $\log\varphi$ and

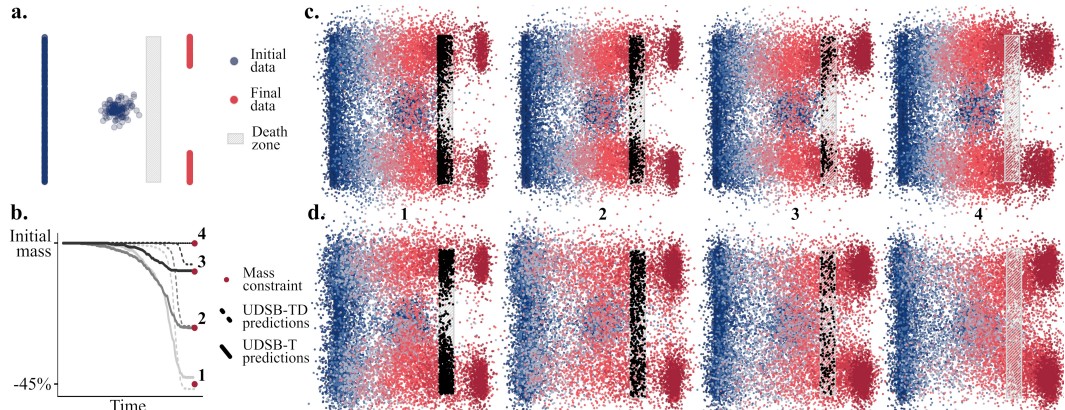

Figure 5: Comparison between UDSB-TD and UDSB-F. (**a**) Points, initially distributed according to the *blue* marginal, move towards the *red* one. In doing so, they cross a region with non-zero killing probability (*gray* rectangle). Assuming 4 different amounts of end mass (1-4), we use (**c**) UDSB-TD and (**d**) UDSB-F to compute particle paths. (**b**) The algorithms find similar, but not identical, trajectories, which respect the end mass constraints in all cases. However, the predicted amounts of live particles at intermediate times differ, owing to the different updates of $\Psi$ used: deaths are clustered by UDSB-TD at the top and bottom of the killing zone and appear, instead, more uniformly distributed in (**d**).

| | Algorithm Comparison | | | |
| | MMD $\downarrow$ | | $W_\varepsilon \downarrow$ | |
| Final mass | UDSB-TD | UDSB-F | UDSB-TD | UDSB-F |
| --- | --- | --- | --- | --- |
| 45% | **8.92e-3** | 12.60e-3 | **1.33** | 1.70 |
| 56% | 9.42e-3 | **6.74e-3** | 1.61 | **1.28** |
| 67% | **3.28e-3** | 5.65e-3 | 1.14 | **1.08** |
| 78% | **5.78e-3** | 6.58e-3 | 1.52 | **1.28** |
| 89% | **5.43e-3** | 8.03e-3 | **0.96** | 1.53 |
| 100% | **1.74e-3** | 5.96e-3 | 1.24 | **1.06** |

Table 2: **Prediction quality on toy dataset.** MMD and Wasserstein distance between the predicted end marginal and the ground truth. The two algorithms UDSB-TD and UDSB-F perform comparably, with the former often achieving better MMD scores and the latter faring better with $W_\varepsilon$.

$\log \hat\varphi$ gradually improve, the algorithm readjusts the trajectories to reach the desired amounts of particle deaths. However, the initial steering persists for the particles starting closer to the death zone (central *blue* points in Fig. 5a) since their trajectories underwent the most significant distortion. Thanks to the improved stability of the Ferryman loss, instead, the trajectories computed by UDSB-F evolve more gradually during training, and particles are therefore free to cross the killing zone (and die).

**Cellular dynamics.** We then proceed to compare our algorithms on a real-world dataset involving the treatment of tumor cells with *Ulixertinib*. Unlike the drug considered in §6, Ulixertinib does not induce measurable cell proliferation and its action on tumor cells is, therefore, best modeled by a death-only (forward) diffusion We measure the quality of predicted cell statuses both at intermediate and final times and report our findings in Table 3. As expected, both algorithms outperform the baseline, with UDSB-F achieving the lowest MMD and Wasserstein distances from the ground truth, owing to its improved stability on higher-dimensional data. Lastly, we report that, upon disabling the killing mechanism, the prediction quality of both algorithms worsened, further stressing the benefits of unbalanced SBs.

| Method | Intermediate time (**24h**) | | Final time (**48h**) | |
|---|---|---|---|---|
| | MMD $\downarrow$ | $W_\varepsilon \downarrow$ | MMD $\downarrow$ | $W_\varepsilon \downarrow$ |
| Chen et al. (2021) | 7.06e-2 (0.04e-2) | 7.99 (0.01) | 1.61e-2 (0.02e-2) | 6.03 (0.04) |
| UDSB-TD (Ours) | 5.34e-2 (0.14e-2) | 7.53 (0.08) | 1.64e-2 (0.11e-2) | 5.64 (0.09) |
| UDSB-F (Ours) | **4.93e-2** (0.19e-2) | **7.07** (0.16) | **1.03e-2** (0.05e-2) | **5.49** (0.18) |

Table 3: **Death-only cellular dynamic prediction results.** MMD and Wasserstein distance between the predicted intermediate/end marginals and the ground truth. UDSB-TD and UDSB-F both generally outperform the baseline, with the latter achieving the best score on both metrics.

**G.2 Models and Training for Toy Datasets.** This section contains details about the experiments that involve synthetically generated datasets since all models used in experiments on low-dimensional datasets share similar architectures and training procedures.

The architecture of networks $f^\theta$ and $\hat{f}^{\hat\theta}$ consists of:

- `x_encoder`: a 3-layer MLP, with 32-dimension wide hidden layers, which takes points in the state space as input;
- `t_encoder`: a 3-layer MLP, with 32 hidden dimensions, which takes the sinusoidal embedding (over 16 dimensions) of the time $t$ as input;
- `net`: a 3-layer MLP, with 32-dimension wide hidden layers, which receives the concatenation of the outputs of the two previous modules as input and outputs the score.

All the above networks use the `SiLU` activation function.

We parametrize $g^\zeta$ with a 5-layer MLP with 64 hidden dimensions and `Leaky-ReLU` as non-linearity.

We run 15 iterations of our unbalanced IPF algorithm and update weights using the ADAMW optimizer with gradient clipping. We set the initial learning rate to 1e-3 for $f$ and $\hat{f}$, and to 1e-2 for $q$.

**G.3 Cell Drug Response.** We further comment on the pre-processing of the cell evolution dataset and provide details on the model architecture and training procedure.

G.3.1 DATASET

We start with the dataset collected by Bunne et al. (2021), which captures the temporal evolution of melanoma cells treated with a mix of the cancer drugs *Trametinib* and *Erlotinib*. The drug is given at time $t = 0$, and the first measurement, which happens at time $t = 8h$, examines a population of 2452 cells. Each cell is characterized by 78 features which are a combination of *morphological* features derived from microscopy (such as cell shapes) and detailed information on the abundance and location of proteins —obtained via the powerful Iterative Indirect Immunofluorescence Imaging (4i) technique (Bunne et al., 2021)– and which we refer to as *intensity* features. Intensity features can be, in turn, categorized as measuring either the sum or average intensity. We remove the 28 features belonging to the former group and are, therefore, left with a 50-dimensional state space. We split the dataset into training and test sets according to an 80/20 split.

Two subsequent measurements of cells in the population take place at times $t = 24h$ and $t = 48h$. Each captures a different population, owing to cell death and birth. In particular (see Fig. 4d), the population is found to have grown by 35% after 24h while shrinking down to 75% of its original size at time $t = 48h$.

To compute the unbalanced DSB in Fig. 4e, we assume a Brownian prior motion of cells in the feature space. Furthermore, we consider a prior killing rate proportional to the distance from the first-order spline interpolation ($g$) of the empirical means of the train set. More precisely, this killing function penalizes cell statuses that deviate substantially ($> 2\sigma$) from $g$ in more than 20% of features. The birth rate is, instead, proportional to a Gaussian Kernel Density Estimation (KDE) computed on the train set.

| Method | MSE $\downarrow$ |
|---|---|
| Spline interpolation (past) | 3.07e-1 |
| Spline interpolation (endpoints) | 1.60e-1 |
| SIR model | 2.54e-1 |
| UDSB-F | **1.12e-1** |

Table 4: **Prediction quality on COVID dataset.** MSE between the predicted and observed presence of Delta variant cases in Europe. UDSB-F beats the baselines consisting of a SIR infection model and of two cubic spline interpolations, using respectively the first 4 measurements (*past*), and the first and last 2 (*endpoints*).

### G.3.2 MODEL AND TRAINING

TThe algorithm UDSB-F requires 3 networks: $f^\theta$ and $\hat{f}^{\hat{\theta}}$ and $g^\zeta$.

The architecture of the first two consists of:

- `x_encoder`: a 3-layer ResNet, with 300-dimension wide hidden layers, which takes the cell status $x$ as input;
- `t_encoder`: a 3-layer MLP, with 32 hidden dimensions, which takes the sinusoidal embedding (over 16 dimensions) of the time $t$ as input;
- `net`: a 3-layer MLP, with 300-dimension wide hidden layers, which receives the concatenation of the outputs of the two previous modules as input and outputs the score.

All the above networks use the `SiLU` activation function.

To parametrize $g^\zeta$, instead, we use a 5-layer MLP with 64 hidden dimensions and `Leaky-ReLU` as non-linearity.

We run 10 iterations of our unbalanced IPF algorithm and update weights using the ADAMW optimizer with gradient clipping. We set the initial learning rate to 1e-3 for $f$ and $\hat{f}$, and to 1e-2 for $q$. We use a batch size of 512.

**G.4 COVID Variants Spread.** To further test unbalanced SBs on real-world phenomena, we model the global evolution of the COVID pandemic over a 4-month period. We aim to reconstruct how multiple COVID variants propagate across countries between April 5, 2021, and August 9, 2021. We choose this time window because it encompasses the appearance and quick spread of the Delta variant, at the expense of the once-dominant Alpha variant.

It is important to emphasize that our task focuses on reconstructing the historical trajectory of the pandemic based on its known initial and final statuses. This differs from the more common practice (Cao and Liu, 2021; Nixon et al., 2022) of predicting how viruses spread solely based on present epidemiological data. Although our approach may not be directly applicable to guiding policies and public health responses during an ongoing outbreak, it can nonetheless provide valuable insights into the transmission and mutation of a pathogen, when its evolution at intermediate times is not known.

**Results.** By running our algorithm UDSB-F on two snapshots of the COVID pandemic, we are able to model it at intermediate times. We can reconstruct the change in variant prevalence both in time and space. We can, for instance, successfully model the proliferation of cases of Delta variant in Europe: Fig. 6g plots the predictions for each variant against the ground truth, which is only known by our algorithm at the initial and final times. Remarkably, UDSB-F reconstructs the spread of the Delta variant better than the baselines (Fig. 6h) and achieves the smallest Mean Square Error (MSE) with respect to the observations (Table 4). Besides offering variant counts for each continent, our predictions also provide time-resolved density estimates of COVID cases for each country (Fig. 6j).

We now provide further details regarding the dataset and how we represent the location and variant of COVID cases.

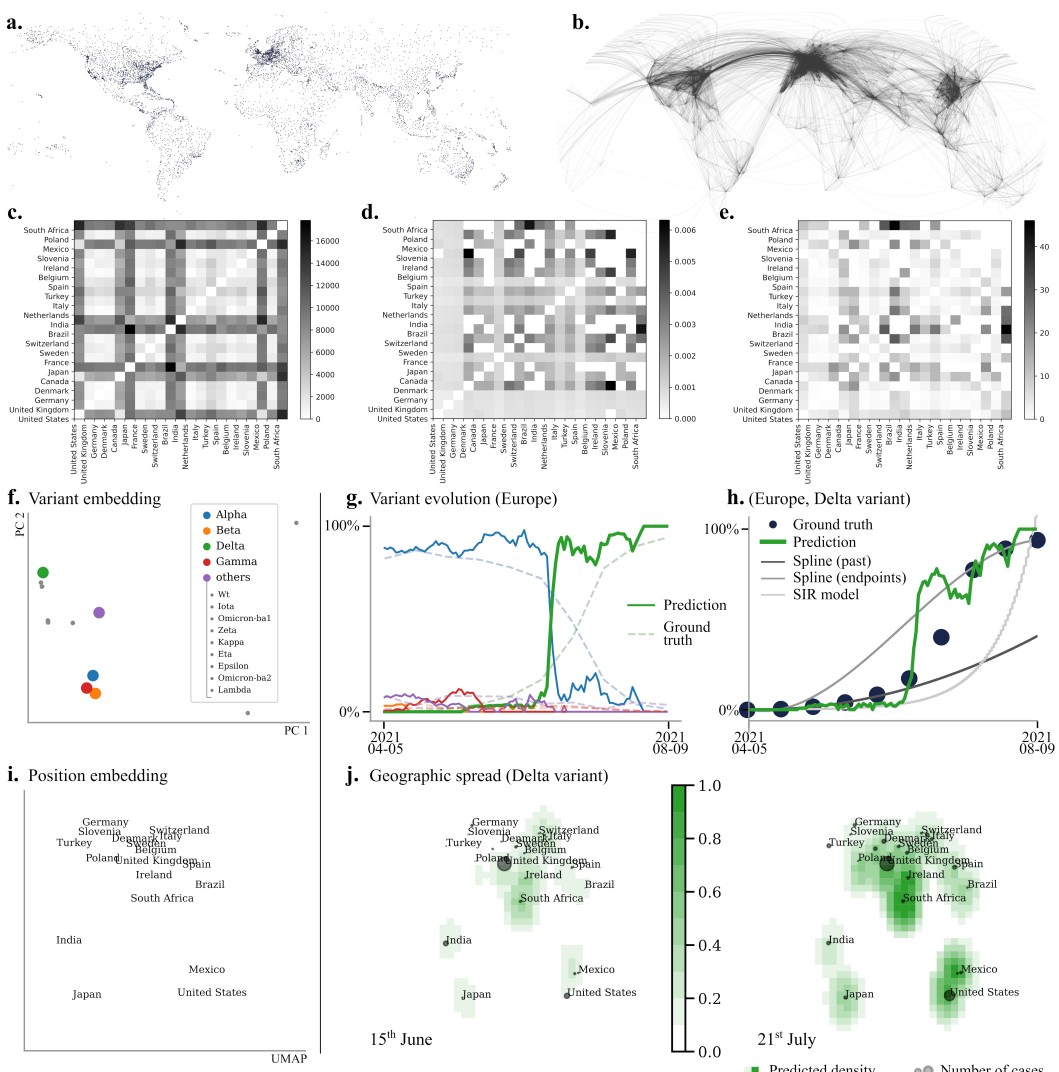

Figure 6: **COVID evolution experiment**. We model variants spreading across countries, by diffusing over a state space $\mathcal{X}$ endowed with location- and variant-dependent information. We represent (**i**) the location of COVID cases, by leveraging the notion of *effective* distance between countries. It is based on (**b**) the number of flights that connect airports (**a**) in different countries. We compute (**e**) *effective* distances as the product of (**c**) *flight* distances $D^F$ and (**d**) *geographic* distances: for the sake of clarity, we only plot pairwise between the 21 countries with the highest number of COVID cases in 2021. (**f**) We encode different COVID variants by means of low-dimensional representations of their spike proteins obtained from Protein Language Models. (**g**) By running UDSB-F on $\mathcal{X}$, we can recover the evolution of COVID variants in Europe more accurately than the baselines (**h**). (**j**) Furthermore, we approximately reconstruct the geographic distribution of cases around the globe at intermediate times.

### G.4.1 DATASET

We start from bi-weekly records[2] of sequenced viral samples around the world. They provide the number of cases linked to different COVID variants which are found in each country on a given day. We only consider samples belonging to the 4 most prevalent variants in our time range, i.e., *Alpha*, *Beta*, *Gamma*, and *Delta*, and represent all the other variant with a fifth category, named *others*. Each

---

[2]Dataset available at: https://www.kaggle.com/datasets/gpreda/covid19-variants

of the datapoints referring to the period between 2021-04-05 and 2021-08-09 therefore contains the following entries:

- `location`: country for which the variants information is provided;
- `date`: the date of the measurement;
- `[variant]`: fraction of sequences belonging to the given variant;
- `num_sequences`: number of sequences processed in the country.

When training our model, we only use the first and last measurements and leave the other ones for testing.

We represent COVID cases that spread among countries as vectors $\mathbf{c}^T = (\mathbf{x}^T; \mathbf{v}^T) \in \mathcal{X}$, in which the first block of coordinates ($\mathbf{x}$) describes the geographic location of the virus and the second one ($\mathbf{v}$) identifies its variant. The use of standard SBs would require that points move *continuously* in $\mathcal{X}$. While we can choose both kinds of embedding to make this assumption approximately true, it is hard to strictly enforce it. This issue emphasizes the need for USBs, which accounts for jumps in the diffusion. In this case, jumps allow (i) new variants to appear in locations where they were initially not present and (ii) local outbreaks to suddenly disappear.

**Location embedding.** To (approximately) respect the continuity hypothesis, we cannot directly use country coordinates to encode the location $\mathbf{x}$ of viruses. In fact, COVID cases do not preferentially propagate to neighboring countries, and using this representation would then render SBs a poor fit to model such dynamics. We, therefore, set to leverage the intuition that the virus tends to spread among countries that are either (i) close to each other or (ii) well-connected, especially via air transportation.

Inspired by the approach pioneered by Brockmann and Helbing (2013), we compute an *effective* distance between countries, which better represents the ease with which COVID moves between them.

To compute effective distances, we rely on flight data, obtained by OpenFlights[3]. Starting with 66934 commercial routes (Fig. 6b), between 7698 airports (Fig. 6a), we count the number of flights $C_{ij}$ linking airports of country $i$ with those in country $j$. We first compute the *flight distance* $D_{ij}^F$ between $i$ and $j$ as $D_{ij}^F = 1/(C_{ij} + \epsilon)$ (where $\epsilon$ is a small constant) (Fig. 6c). In words, a small flight distance implies the existence of frequent connections between two countries. By multiplying flight distances $D_{ij}^F$ by the geographic distance between countries (Fig. 6d), we obtain the matrix $D^E = (D_{ij}^E)_{ij}$ of *effective* distances (Fig. 6e). $D^E$ is a connectivity matrix on the graph of nations and we can measure the shortest paths ($S_{ij}$) between nodes. We then embed in 2 dimensions the manifold induced by the metric $(S_{ij})_{ij}$ via UMAP (McInnes et al., 2020) and therefore obtain a planar representation of the *effective* positions of countries (Fig. 6i). This layout resembles the world map, e.g., it clusters European nations at the top, but also reflects the ease of traveling between countries, e.g., India is equally far from the US and from the UK.

**Variant embedding.** Having described how we determine the location coordinates ($\mathbf{x}$), we now turn to the representation of COVID variants ($\mathbf{v}$). We aim to capture the biological similarity between variants based on the similarity of corresponding Spike proteins, which are critical in determining their infectiousness.

We focus on the mutations occurring in the receptor-binding domains (RBDs), a key part of the Spike protein. More specifically, we encode the amino-acid sequences describing RBDs using ESM-2 (Lin et al., 2023), a state-of-the-art language model optimized for proteins. When projected along their biggest 2 principal components, these embeddings organize in the plane as shown in Fig. 6f. We construct the representation of the *other* variant type, by averaging the embeddings of all the variants excluded by our analysis (*gray* dots in Fig. 6f). As expected, the Beta variant shares more similarities with Alpha than with Delta, which emerged later in time and features additional mutations in its RBD.

**Prior death/birth rates.** To complete the description of the SB problem at hand, we need to specify the prior killing and birth functions. We assume a positive prior probability that Delta cases emerge in

---

[3]https://openflights.org

India since early Delta samples were detected in that country. Furthermore, we introduce birthplaces (for all variants) close to every country and killing zones everywhere else. This prior allows UDSB-F to strike a balance between spreading the virus by making it move across countries (initial infections) and growing native clusters (local proliferation).

### G.4.2 MODEL AND TRAINING

We require 3 networks to parametrize the forward and backward scores ($f$ and $\hat{f}$) and the multiplicative reweighting factor of the posterior killing function ($g$).

We use the same model architecture for $f^\theta$ and $\hat{f}^{\hat\theta}$, which consists of:

- `x_encoder`: a 3-layer MLP, with 64-dimensional hidden layers, which takes $c \in \mathcal{X}$ as input;
- `t_encoder`: a 3-layer MLP, with 32 hidden dimensions, which takes the sinusoidal embedding (over 16 dimensions) of the time $t$ as input;
- `net`: a 5-layer MLP, with 64-dimension wide hidden layers, which receives the concatenation of the outputs of the two previous modules as input and outputs the score.

All the above networks use the `SiLU` activation function.

To parametrize $g^\zeta$, instead, we use a 5-layer MLP with 64 hidden dimensions and `Leaky-ReLU` as non-linearity.

We run UDSB-F for 10 iterations with a batch size of 1024 and update the weights using the `AdamW` optimizer, with initial learning rates of 1e-3, for $f$ and $\hat{f}$, and 1e-2 for $g^\zeta$.

**Variant assignment.** Given a point $\mathbf{c}^T = (\mathbf{x}^T, \mathbf{v}^T)$ in the statespace $\mathcal{X}$, we map it to one of the five COVID variants by using a 1-Neighbor Classifier on $\mathbf{v}$, from the `scikit-learn` library (Pedregosa et al., 2011), trained on the representations in Fig. 6f.

**Baselines.** To assess the performance of our method, we first restrict to the sub-task of predicting how the Delta variant spreads in Europe (Fig. 6h), in the time period under consideration. We consider the following 3 different baselines:

- Spline interpolation (*past*): a cubic spline interpolation of the first 4 available measurements, i.e., those happening between April 5, 2021 and May 17, 2021. In this period, the Delta variant was not yet widespread in Europe, and, as a consequence, the fitted curve underestimates the number of COVID registered in later months.
- Spline interpolation (*endpoints*): a cubic spline interpolation of the first 2 and last 2 known measurements.
- SIR model: a popular ODE-based model (Kermack and McKendrick, 1927) of infectious diseases.

**G.5 Evaluation Metrics.** In this section, we detail the evaluation metrics used to benchmark our algorithms.

**Wasserstein-2 distance.** We measure accuracy of the predicted target population $\hat{\nu}$ to the observed target population $\nu$ using the entropy-regularized Wasserstein distance (Cuturi, 2013) provided in the `OTT` library (Bradbury et al., 2018; Cuturi et al., 2022).

**Maximum mean discrepancy.** Kernel maximum mean discrepancy (Gretton et al., 2012) is another metric to measure distances between distributions, i.e., in our case between predicted population $\hat{\nu}$ and observed one $\nu$. Given two random variables $x$ and $y$ with distributions $\hat{\nu}$ and $\nu$, and a kernel function $\omega$, Gretton et al. (2012) define the squared MMD as:

$$\text{MMD}(\hat{\nu}, \nu; \omega) = \mathbb{E}_{x,x'}[\omega(x, x')] + \mathbb{E}_{y,y'}[\omega(y, y')] - 2\mathbb{E}_{x,y}[\omega(x, y)].$$

We report an unbiased estimate of $\text{MMD}(\hat{\nu}, \nu)$, in which the expectations are evaluated by averages over the population particles in each set. We utilize the RBF kernel, and as is usually done, report the MMD as an average over the length scales: $2, 1, 0.5, 0.1, 0.01, 0.005$.

