# OpenReview forum: "Unbalanced Diffusion Schrödinger Bridge"
_ICLR.cc/2024/Conference — Submitted to ICLR 2024_

### Official Review · Reviewer_1Zr1 · 2023-10-25

**Soundness:** 2 fair
**Presentation:** 2 fair
**Contribution:** 3 good
**Rating:** 5
**Confidence:** 3

**Summary:**

This paper studies the extension of the Schrödinger Bridge problem to the unbalanced setting where two measures of different mass can be considered. To solve the Unbalanced Schrödinger Bridge problem, the authors want to extend the Iterative Proportional Fitting (IPF) procedure and discretize it. To do so, they derive a time-reversal formula with killing and birth terms which are used to adapt the IPF algorithm to the defined problem. Then they explain how to sample from their algorithm and empirically study it on synthetic and single-cell dynamic datasets.

**Strengths:**

i)The introduction and the motivations are clear.

ii) The authors provide an extensive theoretical framework of their proposed method. They first derived an optimization problem and then studied the optimality conditions and then the algorithm to solve it. Finally, they explain how to sample from it.

iii) The authors show how their method performs on a synthetic and a real-world dataset.

**Weaknesses:**

i) The paper is mathematically heavy and I find its clarity poor. A lot of important mathematical quantities are assumed to be known by the reader as well as some of their properties. For instance:
1. The generator (Eq. 3) is not derived naturally and it is hard to understand what its purpose is at first.
2. The decomposition of the generator K is briefly mentioned in Section 4 and the properties of the decomposed parts are assumed to be understood
3. The DSB method [1] is described rapidly in the related work and it is hard to make the connections between the balanced and unbalanced cases on some concepts
4. Generally because of the lack of presentation of the generator and the brief introduction on DSB, I find it hard to follow and understand the extension of the IPF algorithm (ie the full Section 4).

ii) I find the experiments not extensive enough. The authors only compared their method with one competitor. They could have also considered the following Schrödinger Bridge problem approaches (that deal with the balanced case) [1,2,3] and if they cannot be applied to this dataset, they could have considered other single-cell datasets [4].

[1] Diffusion Schrödinger Bridge with Applications to Score-Based Generative Modeling, De Bortoli et al.
[2] Diffusion Schrödinger Bridge Matching, Shi et al.
[3] Simulation-free Schrödinger bridges via score and flow matching, Tong et al
[4] TrajectoryNet: A Dynamic Optimal Transport Network for Modeling Cellular Dynamics: Tong et al.

**Questions:**

What happens when the measures have the same mass? Do you recover the original IPF and Schrödinger Bridge?

Can your method be applied to other single-cell dynamics datasets such as [4]? If so, how does your method compare to other methods?

I think the submission's clarity could be improved a lot by giving more details on the Diffusion Schrödinger Bridge in the related work and linking the different concepts to their unbalanced extension in Sections 4 and 5. Currently, the knowledge and understanding of the DSB method are assumed to the reader, making it hard for non-experts to understand the paper and its theory.

Due to the lack of clarity and the lack of competitors and datasets, I feel that this paper should go under a major revision to be accepted at ICLR. Therefore, my current vote is to reject the paper. While I find the theory to be strong and interesting, it might be possible that the paper's scope is too theoretical for a machine-learning venue like ICLR.

###################
Edit post rebuttal:
###################
Thank you for the rebuttal. I agree that this is a mathematical paper. In my opinion, it deserves a major rewriting to make it accessible to a larger audience and to non-experts in order for them to understand how to apply the proposed method to real-world problems.

Thank you for the novel experiments. I still think that it would be valuable to the paper to consider other single-cell datasets like [4].

Due to the mentioned weaknesses above, I keep my current score.

---

> ### Author Response · Authors · 2023-11-23
> **Reply to reviewer 1Zr1**
>
> Thank you for the time invested in reviewing our paper.
>
> > **The paper is mathematically heavy and I find its clarity poor. A lot of important mathematical quantities are assumed to be known by the reader as well as some of their properties. [...] While I find the theory to be strong and interesting, it might be possible that the paper's scope is too theoretical for a machine-learning venue like ICLR.**
>
> We are aware that our paper includes many mathematical details that can confuse – rather than inform – the reader and ultimately conceal the relevance of the content.
>
> We have nonetheless chosen to submit our manuscript to ICLR in order to highlight the applicability of our proposed method in practical applications, such as those in cancer biology. Our submission reflects the bigger trend involving SBs – long studied theoretically but only recently coupled with efficient solvers – which are now becoming a tool to apply in real-world scenarios.
>
> > **I find the experiments not extensive enough. The authors only compared their method with one competitor. They could have also considered the following Schrödinger Bridge problem approaches (that deal with the balanced case) [4, 5, 6] [...].**
>
> Even though we explicitly mention one baseline in our manuscript, we effectively also compare our results against [8], since the latter coincides with UDSB, after disabling its birth/death mechanism.
>
> We report the performance of this method (e.g., in Table 1) with the label _Ours, no births/deaths_. We explicitly reference Liu et al. [8] only in the caption since we use our own implementation (which coincides with our UDSB code, just without killing function), rather than the one published by the authors. We do this to highlight that the performance improvement cannot be simply explained by differences in implementation (e.g., training, optimizers, …) but must be linked to the killing mechanism.
>
> Moreover, we have compared our algorithms also against De Bortoli et al. [4], but did not report the results because we encountered convergence issues and eventually got worse-quality predictions, compared to [10]. We nonetheless provide our measurements below:
>
> |                                                                      |              **MMD**               | $\textbf{W}_\varepsilon$  |
> |:--------------------------------------|:---------------------------:|:--------------------------------:|
> | De Bortoli et al. [6]                           |       12.8e-2 (0.07e-2)     |                10.53 (0.02)                |
> | Ours                                                          |  **1.13e-2** (0.07e-2)  |              **5.52** (0.09)            |
>
>
> > **What happens when the measures have the same mass? Do you recover the original IPF and Schrödinger Bridge?**
>
> Yes, that is the case, since when the mass at both marginals is equal, $\Psi = 0$ (this can be seen from eq. 33). As a consequence, the jump-related terms in infinitesimal generators disappear and all the processes revert to standard diffusions in $\mathbb{R}^D$. We will more explicitly state that our method is a (strict) generalization of the standard SB problem in the manuscript.

---

### Official Review · Reviewer_wpKe · 2023-10-31

**Soundness:** 3 good
**Presentation:** 3 good
**Contribution:** 2 fair
**Rating:** 5
**Confidence:** 1

**Summary:**

The main contribution of this paper is to extend the Diffusion Schrodinger Bridge (DSB) to the unbalanced case, by introducing time reversal of birth and death of measure in the process. Based on the time-reversal, the authors introduced the iterative proportional fitting scheme, the sampling scheme and finally the entire algorithm

**Strengths:**

This paper is written clearly and mathematically sound. The conditions and statements of every theorem are written rigorously and the logic between different sections are also flawless. It is good for readers to follow the math.

**Weaknesses:**

Disclaimer: I am not familiar with Schrodinger bridge, let alone Diffusion Schrodinger bridge, so my comments might be biased. Please ignore my questions If the authors or the chair found my questions too naive.

I'm curious about what are the uniqueness of SB/DSB/UDSB methods? Is there any real applications where SB/DSB/UDSB works well and other methods does not work? The authors conducted one synthetic experiment and one cellular dynamic experiments comparing with other works. It is good for those who familiar with previous SB/DSB works, but I believe most of the ICLR audiences are not familiar with that. So it will be good for the authors to compare with some non-SB/DSB methods which are more well-known, in order to prove the advantage of SB/DSB/UDSB methods.

**Questions:**

Same as "weaknesses" section

---

> ### Author Response · Authors · 2023-11-23
> **Reply to reviewer wpKe**
>
> Thank you for your review.
>
> > **[...] I believe most of the ICLR audiences are not familiar with that.**
>
> We agree that we did not sufficiently stress the fundamental differences between flavors of SBs and, instead, directly delved into the details of our proposed method. Below, we include a short survey of SBs for the benefit of the reviewer. We will also take care of clearly defining these terms in the introduction of our manuscript.
>
> > **I'm curious about the uniqueness of SB/DSB/UDSB methods? Are there any real applications where SB/DSB/UDSB works well and other methods do not work?**
>
> Different terms related to Schrödinger Bridges:
>
> - **SBs** are Stochastic Differential Equations (SDEs) which describe the most likely evolution of a system with known initial and final states. When assuming that the members of a population (in the original example, the particles of a gas) evolve continuously in time, it turns out to be possible to recover the dynamics of the system by solving (coupled) SDEs for which no analytical solution is known in the general case. This means that, although conceptually very powerful, the SB framework by itself does not provide an easy way to compute the evolution of a real-world system for which (i) the state space is not low-dimensional or (ii) the initial/final distributions are only known empirically.
>
> - **DSBs** are one (very successful) attempt at solving SBs numerically. They heavily rely on the (somehow unintuitive) observation that, as originally proved by the physicist Schrödinger, diffusion processes are fundamentally invertible, i.e., they are still diffusion processes even if the time is played backward. Mathematically, this realization justifies the use of an iterative algorithm, similar to the Sinkhorn algorithm, to compute SB solutions.
>
> - Although computationally sound, DSBs require the (sometimes limiting) hypothesis of mass conservation. In other words, solving DSBs means _‘matching’_ two populations (the initial population, time $t=0$, to the final one, time $t=1$) under the assumption that particles change but none of them disappears. This requirement is often violated in practice (since living beings die and reproduce, and natural systems are rarely isolated from their environments) and this is the fundamental motivation of our work. We show that **UDSBs** (properly) extend DSBs to the case of loss/gain of mass, while still being efficiently solvable via a Sinkhorn-style algorithm, grounded in the theory presented in Sec. 4.
>
> The difference in quality between **UDSB** and **standard SB** solutions rests on the nature of the open system under consideration.
> In particular, we find that if
>
> - the amount of mass exchanged between the system and its environment is non-negligible,
> - additional information about constraints or physically unattainable states is available
>
> then UDSB-F learns dynamics that are  _more meaningful_ than the vanilla SB ones. In fact, when the interaction system/environment can be encoded using death/birth zones, which do not find satisfying equivalents in standard SBs, our experiments point to improved modeling of the phenomenon itself (as opposed, e.g., to the mere refinement of training procedures).

---

### Official Review · Reviewer_z1e1 · 2023-11-01

**Soundness:** 3 good
**Presentation:** 2 fair
**Contribution:** 4 excellent
**Rating:** 5
**Confidence:** 3

**Summary:**

The authors introduce unbalanced diffusion schrodinger bridges a generalization of DSB to the case where mass can vary with time. They introduce a theory of DSB with a forward killing and reverse birth SDEs for an unbalanced diffusion schrodinger bridge. This extends Liu et al. 2022 to the unbalanced setting. A heuristic estimation of $\Psi$ is introduced which avoids estimation of $\log \varphi_t$. These two methods are then applied on a toy example and a single-cell trajectory inference task.

**Strengths:**

- Extends DSB theory to the unbalanced SB problem, and presents a theoretically sound numerical method to approximate solutions to this problem. I found the formulation quite appealing overall. To the best of my knowledge this is an original significant theory that will be useful at least in the subfield of learning cell dynamics.
- Attacks a difficult and important problem in cell modelling.
- The theory is to the best of my knowledge correct, and the exposition of the method is fairly clear although the notation is quite heavy.

**Weaknesses:**

- Experimental evidence: Despite presenting a very interesting theory. Results are presented on one small single-cell dataset. It is difficult to tell how and when to apply UDSB given it is only on a single example with what seems like strong prior knowledge on birth and killing rates. Furthermore, there is only a single baseline for the cell dynamics interpolation task when many specialized methods exist for this task. For me this is the single weakest part of the paper. I would like to see additional comparisons to cell dynamics methods, particularly those based on branching SDEs.
- Placement with regard to existing work: I believe the authors missed work on Trajectory Inference [1,2,3]. Comparison to these methods which also account for cell growth and death in an SDE formulation would greatly strengthen this work and provide context on other ways this problem has been addressed.
- UDSB requires known killing and / or birth rates which may limit in practical applications. It seems like these may be difficult to tune in practice, but the freedom to set them is nonetheless quite interesting.
- Some of the experimental setting is quite unclear at least to me (see questions below).

[1] Lavenant, H., Zhang, S., Kim, Y., & Schiebinger, G. (2021). Towards a mathematical theory of trajectory inference.

[2] Lénaïc Chizat, Stephen Zhang, Matthieu Heitz, and Geoffrey Schiebinger. Trajectory Inference via Mean-field Langevin in Path Space. NeurIPS 2022.

[3] Elias Ventre, Aden Forrow, Nitya Gadhiwala, Parijat Chakraborty, Omer Angel, and Geoffrey Schiebinger. Trajectory Inference for a branching SDE model of cell differentiation. 2023.

**Questions:**

- I don’t understand why the reconstruction quality at the end timepoint should be any better. This is an IID generative modeling task. Is DSB underfit here?
- In addition, I don’t understand how the data is partitioned into training and test for this experiment. If the middle timepoint is left out, how is it known that the observed mass increases 35% but then decreases to -25% from the original total mass? The appendix just states a 80/20 test split.
- The requirement that $k$ is non-negative is not a standard requirement in unbalanced dynamic OT. This seems to be necessary in the theory, but is not a requirement in practice. I did not understand why. Could the authors explain this / clarify in the text?
- More of a comment than a question, “simulating virus spread” is mentioned once in the intro, in the conclusion with an extended experiment in the supplement. I think either this should appear in the main text or not be included. I don’t really understand why this is not in the experiments.

----
Post rebuttal Edit:

I thank the authors for taking time to respond with additional clarifications. My concerns remain re: general applicability and experimental validation, therefore my score remain the same.

I think additional validation would improve this work. I suggest the Root dataset from here: https://journals.plos.org/ploscompbiol/article?id=10.1371/journal.pcbi.1009466

as it has outside observations of approximate cell count via imaging.

I think it would be great to show tangible benefits on datasets without this type of prior knowledge too.

---

> ### Author Response · Authors · 2023-11-23
> **Reply to reviewer z1e1 (1/2)**
>
> Thank you for the valuable suggestions provided. Below, we address your comments.
>
> > **Experimental evidence: Despite presenting a very interesting theory. Results are presented on one small single-cell dataset. It is difficult to tell how and when to apply UDSB given it is only on a single example with what seems like strong prior knowledge on birth and killing rates.
> Furthermore, there is only a single baseline for the cell dynamics interpolation task when many specialized methods exist for this task. For me this is the single weakest part of the paper. [...]**
>
> In our experiments, we use the single-cell dataset in [9] because it provides reliable cell counts, which are usually not available in single-cell datasets. In it, the number of cells recorded at every step is, in fact, chosen as a fixed proportion of the total population at that time.
>
> This information is vital for us to prove that our method adjusts births and deaths so as to match the observed evolution in population size, which is the distinctive trait of our contribution. As correctly observed, it is relatively easy to define birth/death mechanisms in this scenario but, as noted below (see answer on virus spread), we can also apply our method on weaker priors.
>
> Furthermore, we would like to stress that we explicitly mention only one baseline in our manuscript, but we effectively compare our results also against [8], since the latter coincides with UDSB, after disabling its birth/death mechanism. We report the performance of this method (e.g., in Table 1) with the label _Ours, no births/deaths_.
> We explicitly reference Liu et al. [8] only in the caption since we use our own implementation (which coincides with our UDSB code, just without killing function), rather than the one published by the authors. We do this to highlight that improvement in performance cannot be simply explained by differences in implementation (e.g., training, optimizers, …) but must be linked to the killing mechanism.
>
> Lastly, we have compared our algorithms also against De Bortoli et al. [4], but did not report the results because we encountered convergence issues and eventually got worse-quality predictions, compared to [10]. We nonetheless provide our measurements below:
>
> |                                                                      |              **MMD**               | $\textbf{W}_\varepsilon$  |
> |:--------------------------------------|:---------------------------:|:--------------------------------:|
> | De Bortoli et al. [6]                           |       12.8e-2 (0.07e-2)     |                10.53 (0.02)                |
> | Ours                                                          |  **1.13e-2** (0.07e-2)  |              **5.52** (0.09)            |
>
>
> > **Placement with regard to existing work: I believe the authors missed work on Trajectory Inference [1,2,3].**
>
> Thank you for pointing this out. We will duly reference these papers in the manuscript. In particular, we will mention that, whenever multiple marginals are available, it is possible to solve a global regularized OT optimization problem which admits (heuristical) extensions to the case of deaths if the growth mechanism is known [2, 1] or a characterizable bias [3] when the latter is unknown. The main difference with respect to our method is that we do not necessarily rely on individual particle states at intermediate times (but only at the initial and final endpoints) and _encode_ instead prior knowledge about the system dynamics as death/birth functions.
>
> > **UDSB requires known killing and / or birth rates which may limit in practical applications. It seems like these may be difficult to tune in practice, but the freedom to set them is nonetheless quite interesting.**
>
> True. UDSBs require _some_ knowledge of the killing mechanism: the more precise its description, the more relevant the use of Unbalanced SBs. However, as we try to show with the aid of the toy dataset, birth (and death) functions do **not** need to mirror the physical (dis-)appearance of mass. They may be used more broadly as a way to include additional information about the nature of the system at hand. For example, in Fig. 3 we use killing functions to compensate for the missing observations in the marginal (empirical) distributions: doing this effectively helps the SB _discart_ the outliers (e.g., the group of particles that is not observed in one of the marginals) and prevent the deterioration of the quality of SB solutions.

---

> > ### Author Response · Authors · 2023-11-23
> > **Reply to reviewer z1e1 (2/2)**
> >
> > > **I don’t understand why the reconstruction quality at the end timepoint should be any better. This is an IID generative modeling task. Is DSB underfit here?**
> >
> > The quality of the final population reconstruction is not _necessarily_ better on every dataset, but should ideally improve if the introduction of the killing mechanism allows for improvement in the trajectories at intermediate times, e.g., by accurately modeling births and deaths or by helping remove outliers. For instance, in our cell dataset, some cells are likely to be killed by the drug (i.e., are underrepresented in the final empirical distribution) while others proliferate (i.e., are overrepresented at $t=1$): the presence of this phenomenon is supported by the fact that, for some drugs, the total population of cells grows over time, signaling that the net balance of births vs. deaths is positive.
> >
> > We argue that the use of standard DSBs in those cases would not be ideal. That is because DSBs operate under the assumption that the evolution of cell statuses is **continuous in the state space**. As such, to accommodate for the presence of ‘outliers’ they need to create artifacts in the reconstructed dynamics, which are more subtle examples of the diagonal trajectories appearing in the DSB solution to our toy example (Fig. 3c and 3d).
> >
> >
> > > **In addition, I don’t understand how the data is partitioned into train and test sets for this experiment. If the middle time point is left out, how is it known that the observed mass increases 35% but then decreases to -25% from the original total mass? The appendix just states a 80/20 test split.**
> >
> > Thanks for the remark, we apologize for not mentioning this clearly and will update the text accordingly. We use the training set to (i) obtain the individual statuses of cells at initial ($t=0$) and final ($t=1$) times and (ii) compute the target mass variation at all time points (including the intermediate one): as mentioned above, we can deduce it from the dataset, by comparing the number of cells analyzed at every time instant.
> >
> > We then let the test cells evolve and benchmark their final distribution to assess the quality of the learned dynamics.
> > Note, however, that we do not feed the individual statuses of cells at the intermediate time to our UDSB algorithm, meaning that our method cannot simply be traced back to some kind of multi-marginal SB.
> >
> > > **The requirement that $k$ is non-negative is not a standard requirement in unbalanced dynamic OT. This seems to be necessary in the theory, but is not a requirement in practice. I did not understand why. Could the authors explain this / clarify in the text?**
> >
> > The non-negativity of $k$ is linked to the fact that, when presenting the unbalanced SB theory, we restrict ourselves to death-only forward processes, i.e., processes in which the total mass is non-increasing in time. The killing function $k \delta t$ is the (spatial) rate of mass variation and it is therefore natural to have $k \geq 0$ in this case. Analogously, we represent births with _negative_ values.
> >
> > > **More of a comment than a question, “simulating virus spread” is mentioned once in the intro, in the conclusion with an extended experiment in the supplement. I think either this should appear in the main text or not be included. I don’t really understand why this is not in the experiments.**
> >
> > We believe that this additional experiment on virus spread is insightful because it shows that birth/death functions do not always need to represent the physical (dis-)appearance of mass but can also be used to reproduce the (soft) transition of population members across categories (in our case, across virus variants). We consider this kind of UDSB application very promising as it allows us to selectively forgo the continuity of particle trajectories in the state space and work with discrete features (if they can be embedded in Euclidean space).
> >
> > We, however, eventually chose to not include this experiment in the main text because we were unable to fairly compare it against existing methods. Our reconstruction task, namely the inference of virus propagation given initial and final known states, is non-standard in epidemiology (which mostly deals with predicting the future course of the disease given its past behavior) and our attempts at surveying the (very) big amount of recent research did not reveal a suitable baseline candidate.
> >
> > We will remove the references to this experiment from the introduction, as they may be confusing for readers, and only mention it in the outlook section.

---

### Official Review · Reviewer_NpkQ · 2023-11-06

**Soundness:** 2 fair
**Presentation:** 2 fair
**Contribution:** 3 good
**Rating:** 3
**Confidence:** 3

**Summary:**

The paper considers a generalisation of the Schrödinger Bridge problem: initial and final marginals of the process are not probability measures. The authors derive that the time-reversal of diffusions with killing terms and show they correspond to diffusions with birth, and vice versa. Based on this result the authors updated the IPF algorithm and demonstrated its performance using biomedical data.

**Strengths:**

- The considered problem statement is important as it naturally generalises the Di usion Schrodinger Bridge problem

- The authors derived some fundamental result about the time-reversal of diffusions with killing terms

**Weaknesses:**

- The experimental results are not very convincing. Even taking std into account still differences between results (for Ours and Ours - no death/births) are not that big. Is this difference really important for a considered downstream task? Ok, MMD is slightly smaller, and so what? Can we really better predict, e.g., responses to cancer drugs in practice?

- Moreover, the dimensionality of considered data is rather limited. Although, the authors consider an important applied problem, it is not clear whether the proposed method is efficient for more high-dimensional tasks

- The authors do not discuss computational complexity and robustness of the method. How does capacity of NNs influence the results?

- The theoretical derivations look OK, but the practical implementation is very difficult to understand, follow and reproduce.

- I have some expertise in diffusion processes, but the text is difficult to follow for non-experts in diffusion processes. Some important propositions, used by the authors while proving their main results, can be provided for completeness in the appendix

- The description of the algorithm and of its each step are difficult to follow in the appendix

**Questions:**

- In Sec. 5, page 6, the authors introduce a TD loss. Why is it needed? Can the algorithm work without it? Any ablation study on how this loss influences the final quality of the results?

---

> ### Author Response · Authors · 2023-11-23
> **Reply to reviewer NpkQ (1/2)**
>
> Thanks for your review and for giving us the chance to elaborate on the relevance and limitations of our work.
>
> > **The experimental results are not very convincing. Even taking std into account still differences between results (for Ours and Ours - no death/births) are not that big. Is this difference really important for a considered downstream task? Ok, MMD is slightly smaller, and so what? Can we really better predict, e.g., responses to cancer drugs in practice?**
>
> The difference in the quality of solutions computed by UDSB-F and standard DSB solvers rests on the nature of the open system under consideration.
>
> In particular, we find that, if
>
>  -  the amount of mass exchanged between the system and its environment is non-negligible and
>  -  additional information about constraints or physically unattainable states is available,
>
> then UDSB-F learns dynamics that are  _more meaningful_ than the vanilla SB ones. In fact, when the interaction system/environment can be encoded using death/birth zones, a scenario that has no counterpart in standard SBs, our experiments point to better modeling of the phenomenon itself (as opposed, e.g., to the mere refinement of training procedures).
>
> More importantly, besides being (modestly) visible in better distributional metrics (e.g., in MMD), the improvement consists in building SB dynamics that look _more similar_ to the natural phenomenon itself. This helps reconcile practical applications of SBs with the spirit of Schrödinger’s original idea: that of faithfully modeling the probable evolution of a _physical_ entity (which evolves according to well-defined admissible behaviors).
>
> > **Moreover, the dimensionality of considered data is rather limited. Although the authors consider an important applied problem, it is not clear whether the proposed method is efficient for more high-dimensional tasks**
>
> Even though we did not include examples of Unbalanced SB solutions on high-dimensional data, we would like to stress that UDSB-F scales roughly like DSB [4]. That is because the two algorithms share very similar structures, with the only difference being that our method also relies on the Ferryman loss (which is conceived to scale well).
>
> A similar observation, however, does *not* apply to UDSB-TD. This theory-compliant version of our algorithm becomes, in fact, unstable in high dimensions because it directly parametrizes the Schrödinger Potential ($\hat\varphi$), which is a (scaled) probability density and therefore decreases exponentially in magnitude upon increasing the dimensionality of the state space.
>
> > **The authors do not discuss computational complexity and robustness of the method. How does the capacity of NNs influence the results?**
>
> Thanks for pointing this out. We should emphasize that, while our theory-compliant algorithm UDSB-TD is significantly slower than balanced SB solvers —because of the TD loss —UDSB-F is not. At its core, UDSB-F consists of gradient updates determined by the MM loss (like in [4]) and by the Ferryman loss, which has a simple form.
>
> Unlike UDSB-TD, it directly parametrizes the scores ($\nabla \varphi$ and $\nabla \hat\varphi$), rather than the Schrödinger Potentials ($\varphi$ and $\hat\varphi$), hence avoiding to differentiate through the network at every score evaluation.
>
> In fact, the improvement of the performances of UDSB-TD was, along with its poor behavior in high dimensions, one of the motivations driving the inclusion of our second algorithm UDSB-F.
>
> > **I have some expertise in diffusion processes, but the text is difficult to follow for non-experts in diffusion processes. Some important propositions, used by the authors while proving their main results, can be provided for completeness in the appendix**
>
> We acknowledge that we rely on a few results from the theory of stochastic processes which are not properly stated in the manuscript. We will properly include them in a new _‘Background’_ section of the appendix.
>
> To further improve its readability, we also kindly ask the reviewer to contribute a list of definitions and results that are currently not properly introduced/recalled.
>
> > **The theoretical derivations look OK, but the practical implementation is very difficult to understand, follow and reproduce.  The description of the algorithm and of its each step are difficult to follow in the appendix**
>
> We acknowledge that it is non-trivial to turn the pseudo-algorithms that we provide into actual code, also due to the notational burden needed to treat forward/backward SDEs.
>
> To make it easier for readers, we plan to release the entire code repository upon publication. We can already offer an anonymized version to the reviewer, which can be downloaded from here: <https://sendanywhe.re/2967NAGS> (the link expires on 25th Nov).
> Our SB solver has been written from scratch, with the aim of following our theory as closely as possible.

---

> > ### Author Response · Authors · 2023-11-23
> > **Reply to reviewer NpkQ (2/2)**
> >
> > > **In Sec. 5, page 6, the authors introduce a TD loss. Why is it needed? Can the algorithm work without it? Any ablation study on how this loss influences the final quality of the results?**
> >
> > As also discussed in [8], the combination of the MM and TD losses is important to learn the Schrödinger densities. Intuitively, the MM loss drives the learning of the scores ($\nabla\log \varphi$ and $\nabla \log \hat{\varphi}$), while the TD removes unwanted terms which may appear as additive factors to the quantity $\log \varphi$ (and $\log \hat{\varphi}$). This is made more precise in [8], particularly in the proof of Proposition 4 (Section A.3.5).
> >
> > Without the TD loss, the MM loss would just constrain the **gradient** of  $\varphi$. This would however not be sufficient in our case, since we use $\varphi$ to compute the normalization quantity $\Psi$.
> >
> > This loss is therefore theoretically _important_ in the sense that it ensures that we solve the true generalization of SBs to unbalanced marginals. However, as noted above, the TD loss has many disadvantages that render it of limited interest in practice. For that reason, we try to emulate its contribution to training by means of the (more tractable) Ferryman loss.
> >
> > Finally, to test whether the addition of the Ferryman loss truly improves the quality of the reconstructed dynamics, we convert our algorithm into the Deep GSB by [8] (a standard SB solver), which is similar in structure to UDSB, upon disabling the death/birth mechanism of the latter. The results – contained, e.g., in Table 1 – show that the good performance of our algorithm (3rd row) with respect to the baseline (1st row) cannot be simply explained by differences in implementation (e.g., training, optimizers, …), since by executing our same code and only disabling the birth/death mechanism (2nd row) we get worse results.

---

### Author Response · Authors · 2023-11-23
**References**

# References

[1] _Lavenant, Hugo, Stephen X. Zhang, Young-Heon Kim and Geoffrey Schiebinger. “Towards a mathematical theory of trajectory inference.” ArXiv abs/2102.09204 (2021)._

[2] _Zhang, Stephen X., Lénaïc Chizat, Matthieu Heitz and Geoffrey Schiebinger. “Trajectory Inference via Mean-field Langevin in Path Space.” ArXiv abs/2205.07146 (2022)._

[3] _Ventre, Elias, Aden Forrow, Nitya Gadhiwala, Parijat Chakraborty, Omer Angel and Geoffrey Schiebinger. “Trajectory inference for a branching SDE model of cell differentiation.” (2023)._

[4] _De Bortoli, Valentin, James Thornton, Jeremy Heng and A. Doucet. “Diffusion Schrödinger Bridge with Applications to Score-Based Generative Modeling.” Neural Information Processing Systems (2021)._

[5] _Shi, Yuyang, Valentin De Bortoli, Andrew Campbell and A. Doucet. “Diffusion Schr\"odinger Bridge Matching.” (2023)._

[6] _Tong, Alexander, Nikolay Malkin, Kilian Fatras, Lazar Atanackovic, Yanlei Zhang, Guillaume Huguet, Guy Wolf and Yoshua Bengio. “Simulation-free Schrödinger bridges via score and flow matching.” ArXiv abs/2307.03672 (2023)._

[7] _Tong, Alexander, Jessie Huang, Guy Wolf, David van Dijk and Smita Krishnaswamy. “TrajectoryNet: A Dynamic Optimal Transport Network for Modeling Cellular Dynamics.” Proceedings of machine learning research 119 (2020): 9526-9536._

[8] _Liu, Guan-Horng, Tian Qi Chen, Oswin So and Evangelos A. Theodorou. “Deep Generalized Schr\"odinger Bridge.” (2022)._

[9] _Lubeck, Frederike, Charlotte Bunne, Gabriele Gut, Jacobo Sarabia del Castillo, Lucas Pelkmans and David Alvarez-Melis. “Neural Unbalanced Optimal Transport via Cycle-Consistent Semi-Couplings.” ArXiv abs/2209.15621 (2022)._

[10] _Chen, T., Liu, G.H. and Theodorou, E.A., 2021. Likelihood training of schr" odinger bridge using forward-backward sdes theory. arXiv preprint arXiv:2110.11291._

---

### Meta-Review · Area_Chair_5gXN · 2023-12-06

**Metareview:**

Given the feedback from all reviewers, I recommend that this paper not be accepted for publication in its current state. While the paper is recognized for its interesting theoretical contributions, significant improvements are necessary in terms of clarity of writing and the scope of numerical evaluations. A unanimous concern among the reviewers is the difficulty in comprehending the key concepts of the proposed method. This issue suggests that the paper currently lacks the clarity required to make its theoretical contributions accessible to a machine learning audience. Therefore, substantial rewriting and clarification are needed to bridge this gap. Additionally, the numerical experiments, although noteworthy (particularly those involving single cell genomics), are not sufficiently comprehensive. The reviewers suggest the need for a broader range of data types and more extensive baseline comparisons. These enhancements are crucial to convincingly support the paper's approach to dealing with the birth and death of particles in a theoretically sound manner. In light of these considerations, it is clear that the paper requires major revisions before it can be considered ready for publication. The current shortcomings, particularly in terms of clarity and the depth of numerical evaluation, are significant barriers to its acceptance.

**Justification For Why Not Higher Score:**

I think the needed rewriting is just too much. It is a pity bc the paper is very strong.

**Justification For Why Not Lower Score:**

N/A

---

### Decision · Program_Chairs · 2024-01-16

Reject